# DNA damage causes rapid accumulation of phosphoinositides for ATR signaling

Yu-Hsiu Wang[1], Anushya Hariharan[1], Giulia Bastianello[2,3], Yusuke Toyama[1,4,5], G.V. Shivashankar[1,2,5], Marco Foiani[2,3] & Michael P. Sheetz[1,6]

Phosphoinositide lipids (PPIs) are enriched in the nucleus and are accumulated at DNA damage sites. Here, we investigate roles of nuclear PPIs in DNA damage response by sequestering specific PPIs with the expression of nuclear-targeted PH domains, which inhibits recruitment of Ataxia telangiectasia and Rad3-related protein (ATR) and reduces activation of Chk1. PPI-binding domains rapidly (<1 s) accumulate at damage sites with local enrichment of PPIs. Accumulation of $PIP_3$ in complex with the nuclear receptor protein, SF1, at damage sites requires phosphorylation by inositol polyphosphate multikinase (IPMK) and promotes nuclear actin assembly that is required for ATR recruitment. Suppressed ATR recruitment/activation is confirmed with latrunculin A and wortmannin treatment as well as IPMK or SF1 depletion. Other DNA repair pathways involving ATM and DNA-PKcs are unaffected by PPI sequestration. Together, these findings reveal that nuclear PPI metabolism mediates an early damage response through the IPMK-dependent pathway to specifically recruit ATR.

[1] Mechanobiology Institute, National University of Singapore, Singapore 117411, Singapore. [2] IFOM, the FIRC Institute of Molecular Oncology, Via Adamello 16-20139 Milan, Italy. [3] Università degli Studi di Milano 20133 Milan, Italy. [4] Temasek Life Sciences Laboratory, 1 Research Link, Singapore 117604, Singapore. [5] Department of Biological Sciences, National University of Singapore, Singapore 117543, Singapore. [6] Department of Biological Sciences, Columbia University, New York, NY 10027, USA. Correspondence and requests for materials should be addressed to M.P.S. (email: ms2001@columbia.edu)

Phosphoinositide lipids constitute a phospholipid family that accounts for only 1% of the total lipid in the plasma membrane. However, evidence from lipidomic mass spectroscopy indicates that the mole fractions of PPIs are nearly 10 times more enriched in the nucleus[1,2], while the phospholipid profiles are conserved in plant and animal nuclei[3,4]. Early evidence that a distinct pool of phosphoinositide exists in the nucleus was established by Smith and Wells[5,6], who demonstrated that phosphatidic acid (PA), phosphatidylinositol 4-phosphate (PI(4)P) and phosphatidylinositol 4,5-bisphosphate (PI(4,5)P2) were rapidly labeled by $^{32}P$ when intact rat liver nuclei were isolated and incubated with $[\gamma\ ^{32}P]$-ATP. This finding indicates that PPI can be synthesized in the nucleus in the absence of cytosolic PPI kinases. How $PIP_2$ regulates nuclear function and in what form PPIs exist in the nucleus has been an open question for decades[7,8].

An earlier study reports that nuclear phosphoinositides are synthesized after DNA damage[9], doubling the level of $[\gamma\ ^{32}P]$-ATP incorporation in nuclear $PIP_2$ within the first hour after ionizing radiation (IR). In addition, cisplatin-induced DNA damage causes an increase in the nuclear $PIP_2$ level through type I PI4P-5K activation, although the PI4P-5K isoform and the mechanism involved remains unclear[10]. Together, these early observations imply that nuclear phosphoinositides have a role in

**Fig. 1** Nuclear $PIP_2/PIP_3$ level increases upon DNA damage induction. **a** Confocal optical slices showing nuclear $PIP_2$ speckles and γH2AX foci in control U2OS cells as revealed by immunostaining before and after being exposed to UV. Secondary antibody labeling $PIP_2$ and γH2AX was conjugated with Alexa Fluor-546 and Alexa Fluor-647, respectively. **b** Corresponding quantification of panel A showing the normalized mean fluorescence intensity of Hoechst 33342, $PIP_2$ and γH2AX within the nucleus. **c** Same anti-$PIP_2$ and anti-γH2AX staining for U2OS cells transfected with 3xNLS-PLCδPH before and after being exposed to UV. **d** Corresponding quantification of panel **c**. Results were normalized to the signal intensities of non-UV treated, non-transfected cells as shown in panel B. Symbols in black indicated $t$-test $p$-values when comparing it to the signal intensity of non-UV treated, non-transfected cells in panel b. Symbols in blue indicated $t$-test results when comparing it to the non-UV treated PLCδPH-expressing cells. Please refer to Supplementary Figure 1 for similar results obtained from cells with chemical-induced DNA damage. **e** Western blots showing suppressed Chk1 activation and increased γH2AX level in MEFs expressing 3xNLS-PLCδPH-EGFP, but not EGFP or cytoplasmic PLCδPH-EGFP upon global UV irradiation. Numbers on the left indicated the molecular weight of the ladder. Cells were exposed to UV for 2 min and harvested after 1hr recovery in an incubator. **f** Corresponding pChk1 ratio with and without being exposed to UV. Scale bar 10 μm. Data are representative of three independent experiments. Error bars represent mean ± s.e.m in all panels. Student's $t$-test, $^*p < 0.05$; $^{**}p < 0.01$; $^{***}p < 0.001$; $^{****}p < 0.0001$. n.s. represents not significant

responding to DNA damage. Therefore, we propose to investigate how nuclear PPIs are regulated and at what stage they contribute to the damage response by examining whether nuclear PPIs are required for recruiting DNA repair

Nuclear PPIs regulate many nuclear functions including transcription[11], splicing[12,38], export[14] of mRNAs, as well as the DNA damage response, which is reviewed elsewhere[11]. However, we don't have a clear understanding of how nuclear PPIs interact with their target proteins. Recently, a protein-PIP$_3$ complex was reported with a nuclear receptor protein, also known as steroidogenic factor (SF1), in which the lipid head group was exposed to enable signaling[15,16]. Since PPI-binding PH domains interact with not only membranous PPIs but also PPIs in complex with nuclear receptors[15], we constructed nuclear localization signal-tagged PH domains to compete for PPI-binding to endogenous interacting proteins in the nucleus. Fluorescent protein-fusions of these NLS-tagged PPI-binding domains not only sequester available PPIs in the nucleus but also serve as reporters for changes in local PPI levels. Although a similar strategy was used to investigate the role of PPIs in membrane-cytoskeleton adhesion[17], these nuclear-targeted domains enable the analysis of the roles of nuclear PPIs, particularly in DNA damage response.

PPI-binding pleckstrin homology (PH) domains were identified in 1993 as a stretch of 100–120 amino acids that appears twice in the platelet protein pleckstrin[18–20] and are the 11th most abundant class of domains in the human genome[21–24]. A whole cell PPI interactome study using human cells revealed 405 PPI-binding proteins with PH domains as the most prevalent[25]. It is important to note that 30% of identified PPI-binding proteins were localized exclusively in the nucleus, while merely 5% of the PPI-binding proteins were associated with the plasma membrane[25]. In another proteomic study, 168 out of 349 nuclear proteins isolated by neomycin extraction contained PPI-binding domains, including PHD, PH and K/R-rich motifs[26]. Compared to the known physiological functions of PPIs in the plasma membrane, very little is known about the role of PPI-protein interactions in the nucleus. There are indications that PPIs could participate in DNA damage responses, since the PIP$_3$-binding Btk-PH domain accumulates at damage sites with a half time around 20 s[27] and several proteins that respond to DNA damage possess a PH domain, such as the p62 subunit of transcription factor II H (TFIIH)[28] and PH-domain only protein (PHLDA3)[29]. Still the roles of nuclear PPIs in DNA damage responses are unclear.

The DNA damage response includes several damage repair mechanisms, including the Ataxia telangiectasia and Rad3-related (ATR), ATM and DNA-dependent protein kinase (DNA-PKcs) pathways. ATR, ATM and DNA-PKcs are essential PI3K-like kinases that mediate DNA repair at the damage site through mechanisms including non-homologous end joining (NHEJ) and homologous recombination repair. The multiple systems for DNA repair are likely to compete with each other to create a robust repair system.

In this study, we confirm that DNA damage by either UV radiation or chemical modification causes an increase in PIP$_2$ synthesis in the nucleus. The roles of PPIs in DNA damage responses are investigated by testing the effect of nuclear PH domain expression on repair protein recruitment. Surprisingly, the recruitment of ATR and ATRIP, but not ATM or DNA-PKcs, is blocked by PH domain expression. Further, localized DNA damage induces a rapid increase in intranuclear PPI levels at the site as revealed by both anti-PIP$_2$ immunofluorescence and recruitment of NLS-tagged PH domains within 1 s of laser microirradiation. Phosphorylation of PIP$_2$ at damage sites is suppressed by IPMK depletion. The pathway from PPI synthesis to ATR recruitment involves actin polymerization similar to the cytoplasmic roles of PPIs[30–32] and consistent with previous

studies of actin in the nucleus[33,34]. This establishes a link between nuclear PIP$_2$, nuclear actin and the DNA damage response through ATR.

## Results

### Nuclear PIP$_2$ level increases upon DNA damage induction.
To determine if earlier studies[9] were relevant in our cell system, DNA damage was generated either chemically or by UV irradiation, and the nuclear level of PIP$_2$ was evaluated by immunofluorescence using an anti-PIP$_2$ monoclonal antibody [clone: 2C11][35,36]. Nuclear PIP$_2$ level of cells sensitized with 10 μg mL$^{-1}$ Hoechst 33342 increased significantly upon exposure to UVC at 1.33 W m$^{-2}$ for 10 min (Fig. 1a, b). The same was true for cells treated with 0.01% MMS (a radiomimetic alkylating reagent that induces DNA strand breaks) for 2 hrs (Supplementary Fig. 1a, b). In both cases, nuclear PIP$_2$ levels remained elevated for the first hour after damage induction. In contrast, the level of γH2AX continued to increase over time. This result suggested that nuclear PIP$_2$ served as an early signal following DNA damage.

In order to understand the role of PPIs in the DNA damage response, we overexpressed PH domains with triple nuclear localization signals (3xNLS) to complex nuclear PPIs and block normal PPI functions. Noticeably, 3xNLS-tagged PH domains showed a clear nucleolar enrichment, which was commonly observed for proteins tagged with SV40 NLS[37] and was independent of the PH domain. In these experiments, 3xNLS-PLCδ-PH domain expression was limited to 12 hrs to avoid potential changes in the protein expression through PPI-dependent pre-mRNA splicing[38] and mRNA export[39,40]. After UVC-induced DNA damage in the presence of 3xNLS-PLCδPH, there was a significant reduction of PIP$_2$ immunostaining (Fig. 1c, d and Supplementary Fig. 1c, d). It suggested that high nuclear concentration of δ domains competed with anti-PIP$_2$ antibodies to reduce PIP$_2$ immunostaining and therefore eliminated UV- and MMS-induced elevation of uncomplexed nuclear PIP$_2$ level as seen in control cells (Fig. 1a, b and Supplementary Fig. 1a, b).

The expression of PLCδPH in the nucleus was genotoxic and resulted in an elevated signal of γH2AX staining even in the absence of exogenous DNA damage (Fig. 1d, e and Supplementary Fig. 1d). With exogenous DNA damage, the γH2AX staining increased over time after damage in both control and PH domain-expressing cells, although, the γH2AX signal was significantly higher in PH domain-expressing cells; hence nuclear PIP$_2$ somewhat preserved genome integrity. Results from western blots showed that PH domain-expressing cells were sensitized to DNA damage as seen by increased γH2AX levels upon UV irradiation (Fig. 1e). However, PH domain expression lowered the increase in Chk1 phosphorylation after UV irradiation (153, 116 and 37% increase for EGFP-, PLCδPH-GFP- and 3xNLS-PLCδPH-EGFP-expressing cells, respectively), hinting that nuclear PIP$_2$ might assist through ATR-mediated signaling (Fig. 1e, f).

### Sequestration of nuclear PPIs suppresses ATR recruitment.
To determine if DNA damage response (DDR) pathways were linked to PPI metabolism, the recruitment of three PIKKs and early DNA damage sensing complexes Ku70-80 and Mre11-Rad50-NBS1 (MRN) at damage sites was compared in the presence or absence of nuclear PPI-binding PH domains. We also examined related components in ATR-dependent pathway such as the recruitment of RPA and ATRIP, and Chk1 phosphorylation.

Initially, three different PPI-binding domains: SidM-P4M, PLCδPH and Btk-PH that bound to PI4P, PIP$_2$ and PIP$_3$, respectively, were tagged with 3xNLS and mCherry (Fig. 2a) and transfected separately into GFP-ATR-expressing cells. After laser

microirradiation of Hoechst-labeled nuclei, the recruitment of GFP-ATR was greatly suppressed (Fig. 2b). Quantification showed that the P4M domain was a weaker inhibitor compared to the PLCδ-PH and Btk-PH domains, indicating that PIP$_2$ and PIP$_3$ played a role in mediating ATR recruitment (see Fig. 2d-f). As a control, expression of cytoplasmic PLCδPH without an NLS

had no effect (Fig. 2c). Importantly, the suppression of ATR recruitment was reversed when a loss-of-function mutation was introduced into the lipid-binding pocket of all three PPI-binding domains. Thus, ATR recruitment to microirradiation-induced DNA damage sites depended upon nuclear phosphoinositides, especially PIP$_2$ and PIP$_3$.

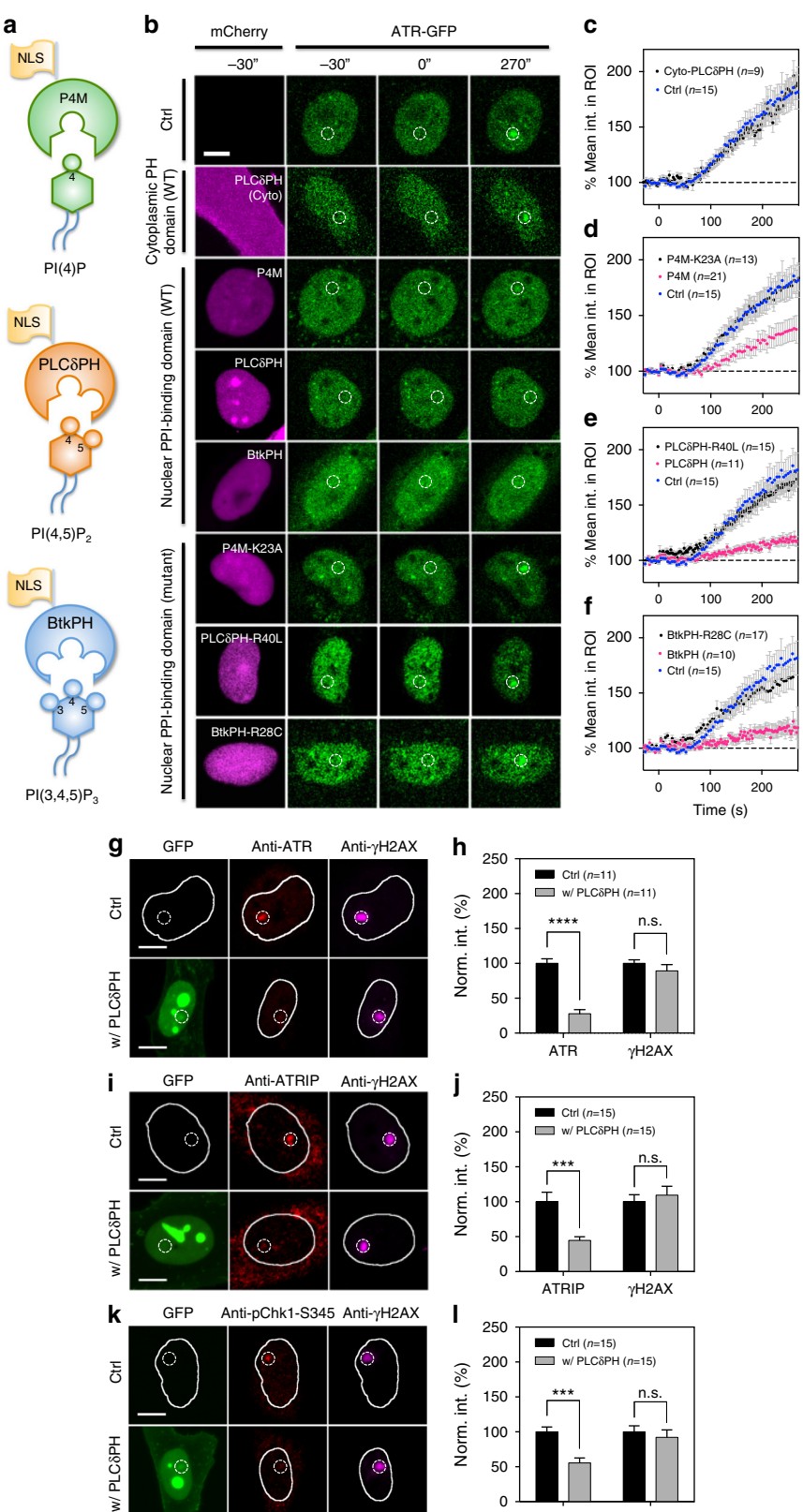

Immunostaining confirmed that endogenous ATR recruitment was suppressed over 70% by the PH domain, while the level of γH2AX staining within the same ROI remained unchanged (Fig. 2g, h). In contrast, the recruitment of the other DNA damage transducers, ATM and DNA-PKcs, was insensitive to nuclear PH domain expression as confirmed by immunostaining (Supplementary Fig. 2a–d). Therefore, it seemed that the ATR-dependent pathway, but not the ATM- or DNA-PKcs-dependent pathway, was affected by nuclear PPI sequestration.

In the canonical ATR signaling pathway, RPA-single-stranded DNA (ssDNA) nucleofilaments mediate the recruitment of ATR and other ATR signaling components to sites of DNA lesion via ATR Interacting Protein (ATRIP). Then, the phosphorylation of Chk1 at Ser345 by ATR served as a downstream signal for ATR activation. Therefore, we measured the recruitment of ATRIP, RPA, and the accumulation of phospho-Chk1. Expression of nuclear PLCδ-PH suppressed both endogenous ATRIP recruitment (Fig. 2 i, j) and pChk1 accumulation (Fig. 2k, l) at damage sites. However, the recruitment and phosphorylation of RPA at Ser4/Ser8 and Ser33 was not affected by PH domain expression (Supplementary Fig. 2e–j). The phosphorylation of RPA at Ser4 and 8 was catalyzed by DNA-PKcs, and was not affected by PH domain expression. However, the fact that the level of pRPA Ser33 phosphorylation was also unaffected indicated that ATR-mediated phosphorylation of RPA at Ser33 might be compensated by other PIKKs[41]. Thus, PPI sequestration prevented recruitment and activation of the ATR-ATRIP-Chk1 signaling axis, despite the presence of RPA at damaged sites.

In addition, we examined the PPI-dependence of the recruitment of Ku70-80 and MRN complexes, as both were considered early components in DDR. However, recruitment of either EGFP-NBS1 (Supplementary Fig. 2k, l) or EGFP-Ku70 (Supplementary Fig. 2m, n) at DNA damage sites was neither decreased nor delayed by the co-expression of PPI-binding PH domains. Thus, only the ATR-dependent pathway of DNA damage repair seemed to depend upon nuclear PPIs.

**Nuclear PPIs actively and rapidly accumulate at damage sites**. Earlier studies from thin-layer chromatography indicated that nuclear $PIP_2$ levels increased after DNA damage by ionizing radiation[9], which could cause PH domain accumulation at DNA damage sites. When the NLS-tagged EGFP-fusion of the PLCδ-PH domain was expressed, the UV laser microirradiation of the Hoechst-sensitized nuclei caused a rapid local loss of fluorescence from photobleaching, followed by an initial rapid recovery due to the diffusion of unbleached molecules and a further increase in the intensity of EGFP above the surrounding background (Supplementary Movie 1). Such rapid accumulation occurred after laser microirradiation of the nucleoplasm (Fig. 3a), but not of the plasma membrane or of the nucleoli. We ruled out the possibility that the accumulation was the result of UV-induced photoconversion of Hoechst 33342 by confirming that there was no detectable increase in green fluorescence with normal photodamage conditions (355 nm UV laser at 1 nW for 500 ms) in the

absence of GFP-fusion proteins. In fact, Hoechst 33342 was chosen because it exhibited the lowest level of UV-induced photoconversion compared with DAPI and Hoechst 33258[42].

The halftime of PLCδPH domain accumulation at the damage site was $0.8 \pm 0.2$ s (mean $\pm$ s.d.), and did not vary significantly with UV laser power. The magnitude of accumulation depended on the PPI-binding construct, cell line used, and the laser power for damage induction. The accumulation magnitude of different PPI-binding domains followed the order: PLC-PH > P4M-SidMx2 > Btk-PH. The increase in the PLCδ-PH domain was 18% at 1 nW, while there was only a minor increase of 6.2% in fluorescence intensity within the ROI in cells expressing EGFP-3xNLS at the same laser power, which was considered a background level of laser-induced protein damage (Fig. 3b). Point mutations in the active site of the PLCδ-PH domain caused a decrease in the level of recruitment to near background levels (Fig. 3c, f). This indicated that there was an increase in PPI density in the damage area after irradiation.

To determine if specific accumulation was due to active, local synthesis of nuclear lipids, we performed a control experiment on nuclear PLCδ-PH-expressing cells killed by the addition of 100 mM β-mercaptoethanolamine (MEA). There was no accumulation of PH domains upon UV microirradiation of the dead cells (Fig. 3d, e). In fact, the fluorescence of PH domains barely recovered at the damage site, and the magnitude of recovery decreased with increasing time of MEA treatment. To test temperature and ATP dependence of the response, we microirradiated cells expressing nuclear PLCδ-PH either at 25 °C or after ATP depletion for 30 min (Fig. 3g). The normalized %Inc values (after subtraction of the control values for EGFP-3xNLS) were reduced by nearly 60% at 25 °C and up to 85% for ATP-depleted cells (Fig. 3h). Together, the results indicated that rapid accumulation of the PH domain after DNA damage was an energy-dependent process that likely reflected the local production of nuclear lipids.

Two additional positive control experiments supported the claim that the rapid accumulation of PH domain with DNA damage was mediated by interaction with PPI in the nucleus. First, we investigated the effect of wortmannin, a potent inhibitor of PI3K on the magnitude of PH domain accumulation. Interestingly, wortmannin caused an increase in $PIP_2$ and a decrease in $PIP_3$ binding PH domain accumulation at damage sites, respectively, (Fig. 3i, j and after background subtraction in Fig. 3k). Secondly, the local enrichment of $PIP_2$ or $PIP_3$ at damage sites was confirmed by colocalization of anti-$PIP_2$ and anti-γH2AX immunofluorescence (Fig. 3l). The major $PIP_2$ foci at the damage sites always correlated with γH2AX staining when DNA damage was introduced by laser microirradiation. However, the minor $PIP_2$ foci didn't always co-localize with γH2AX foci. This was partially explained by nuclear $PIP_2$ involvement in other aspects of nuclear function including pre-mRNA slicing[38] and mRNA export[39,40]. In contrast, the $PIP_2$ and γH2AX foci were highly co-localized when DNA damage was introduced globally using a bench-top UV lamp (Supplementary Fig. 3c–f) and is consistent with earlier findings with ionizing radiation[27]. Further,

**Fig. 2** Sequestering nuclear phosphoinositides suppresses ATR as well as ATRIP and ATR-mediated pChk1 recruitment. **a** Illustration of protein probes used for sequestering specific nuclear PPIs. **b** Confocal montage of ATR recruitment upon laser microirradiation with or without the expression of individual cytosolic or NLS-tagged PPI-binding domains or their loss-of-function mutants. **c** Quantification of ATR recruitment within the ROI with the expression of cytosolic PLCδ-PH. **d-f** Quantification of ATR recruitment within the ROI with or without the expression of SidM-P4M, PLCδ-PH and Btk-PH domain, respectively, or their corresponding mutants. **g** Representative micrographs of control and PLCδ-PH-expressing cells stained with ATR and γH2AX after laser microirradiation and **h** the corresponding quantification of fluorescence signals within the ROI. **i, j** Representative micrographs of similar experiments stained for ATRIP and its corresponding quantification. **k, l** Another similar experiment stained for pChk1-S345 and its corresponding quantification. Dashed circles are 2.5 μm in diameter indicating the ROI for laser microirradiation. Data are representative of three independent experiments. Error bars represent mean $\pm$ s.e.m. Student's $t$-test, ***$p < 0.001$; ****$p < 0.0001$. n.s. represents not significant

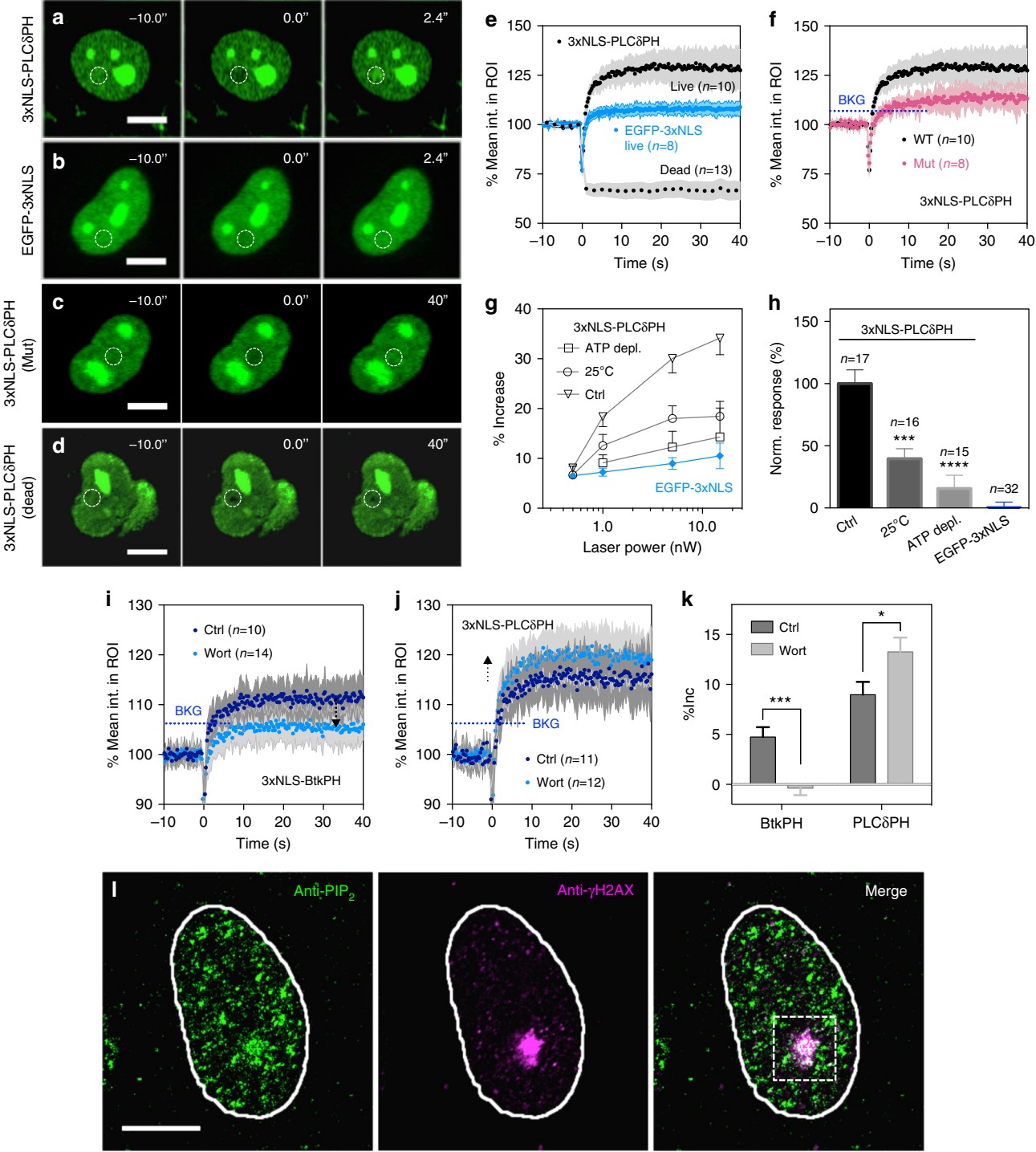

**Fig. 3** Active and rapid accumulation of PH domains upon laser microirradiation is wortmannin-sensitive. Confocal montage of cells after laser microirradiation while expressing **a** 3xNLS-PLCδPH-EGFP, **b** 3xNLS-EGFP, **c** 3xNLS-PLCδPH Mutant (K30L/K32L/R40L) or **d** 3xNLS-PLCδPH-EGFP after cell was killed by addition of 100 mM MEA. **e** Corresponding quantification of the mean fluorescence intensity within ROIs for panels A, B and D. Shaded area represents standard deviation. **f** Quantification of the mean fluorescence intensity within ROIs for panels A and C. Shaded area represents standard deviation. **g** Laser power-dependent measurement of protein accumulation magnitude within ROIs under control, ATP-depleted condition or at a lower temperature. **h** Pooled and normalized magnitude of PLCδ-PH accumulation under different conditions after background subtraction as determined by EGFP-3xNLS. **i, j** The accumulation magnitude of Btk-PH and PLCδ-PH domain, respectively, in the presence or absence of 0.5 μM wortmannin. Blue dotted line indicates the background level determined by EGFP-3xNLS. Error bars represent mean ± s.d. **k** The corresponding quantification of panels **i, j** with background subtraction. **l** Representative immunofluorescent micrographs of cells stained for PIP₂ and γH2AX after laser microirradiation. Laser power for damage induction was set as constant at 1 nW for 500 ms except for panel **g**, **h**. Scale bars are 10 μm for all panels. Dashed circles indicate the ROI for laser microirradiation. Data are representative of three independent experiments. All error bars represent mean ± s.e.m. except panel **i, j**. Student's *t*-test, *$p < 0.05$; ***$p < 0.001$; ****$p < 0.0001$. n.s. represents not significant. BKG represents background level

NLS-tagged PH domains concentrated at PIP$_2$ antibody speckles (Supplementary Fig. 3g–j). Thus, we concluded that the rapid accumulation of PH domains was mediated by local enrichment of PPIs in the nucleus as part of DNA damage responses.

**IPMK depletion suppresses PIP$_3$ accumulation at damage sites.** Since the local accumulation of PH domains was mediated by rapid metabolism of PPIs at damage sites as indicated by wortmannin treatment (Fig. 3i, k), we looked for the PPI kinase(s) involved. Candidate nuclear PPI kinases included type I PI4P-5Kα and β isoforms (PIP5K1A and PIP5K1B), the catalytic unit of Type I PI3Kβ (PIK3CB, also known as p110β) and inositol

polyphosphate multikinase (IPMK). Kinases were either knocked down by RNAi or knocked out by CRISPR-Cas9 genomic editing[43]. Knockout (KO) or knockdown (KD) cells after 66–72 h with the RNAi were transfected with either 3xNLS-PLCδPH-EGFP or 3xNLS-Btk-PH-EGFP for 6 hr. All shRNA constructs were subcloned in a pRFP-C-RS vector. Both PIK3CB KO cells and IPMK KD cells were co-transfected with an additional empty mCherry vector for labeling purposes. Laser-induced accumulation of PIP$_2$- or PIP$_3$-binding PH domains was then compared in both KO/KD and control conditions. Only IPMK depletion significantly suppressed the accumulation of Btk-PH, whereas knocking down other kinases had no effect (Fig. 4a–h).

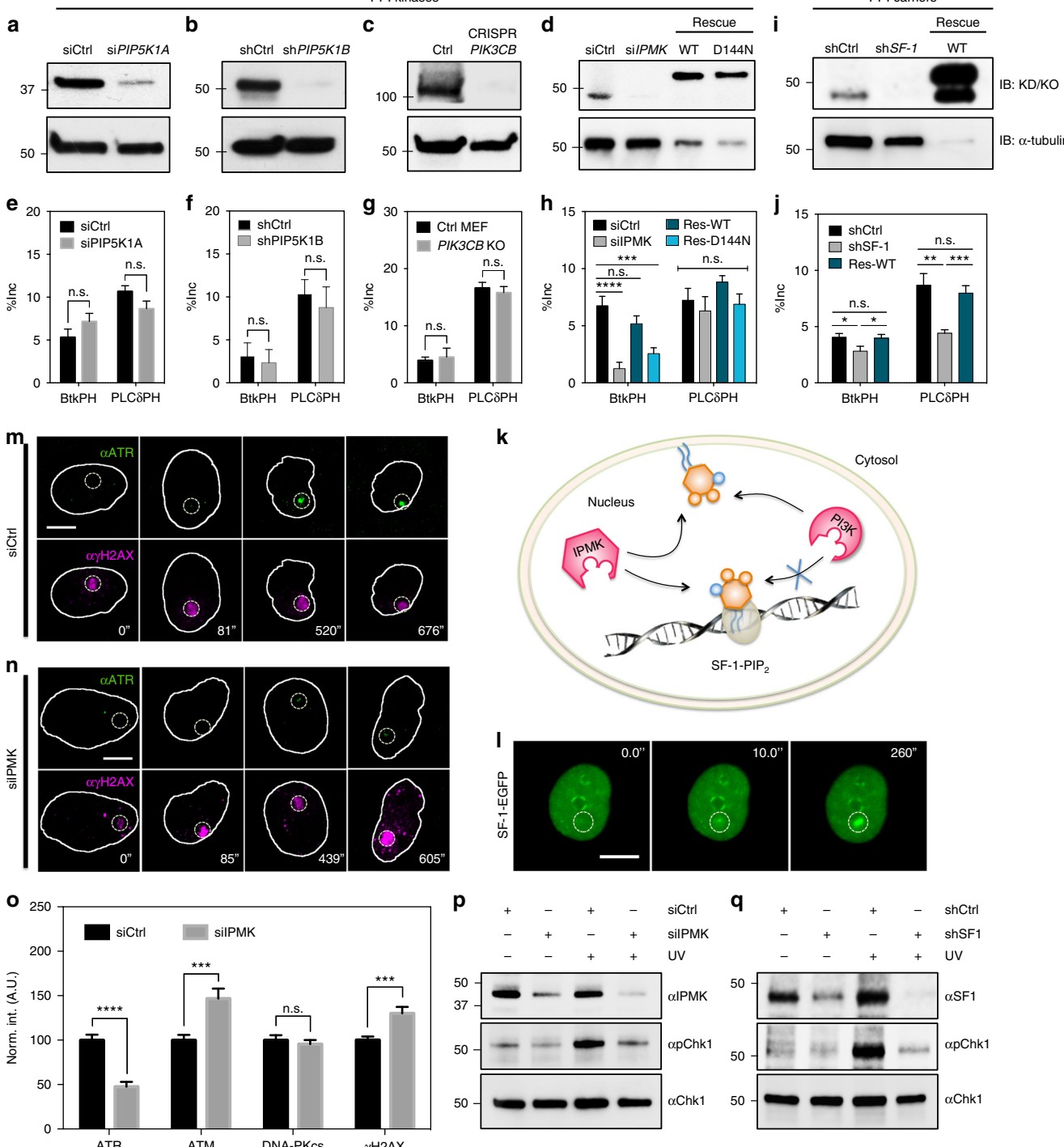

Accumulation of Btk-PH dropped by 80% in IPMK-depleted cells after background subtraction, but PLCδ-PH accumulation showed no significant difference (Fig. 4h). The role of the kinase activity of IPMK was confirmed by transfection with either siRNA-resistant flag-tagged human IPMK wild type (siRes-3xFlag-hIPMK-WT) or kinase dead (siRes-3xFlag-hIPMK-D144N) constructs[40]. Wild type hIPMK restored accumulation of Btk-PH upon laser microirradiation, but the kinase dead mutant did not (Fig. 4h). Thus, phosphorylation of $PIP_2$ to $PIP_3$ depended upon IPMK activity.

Further evidence for IPMK involvement came from the form of $PIP_2$ in the nucleus. Both IPMK and classical PI3K phosphorylated $PIP_2$ in vitro in a lipid mixture or in a membrane structure[44]. However, a distinct pool of $PIP_2$ in the nucleus bound to steroidogenic factor-1 (SF1) in a 1:1 stoichiometric ratio (Fig. 4k). The resulting SF1-$PIP_2$ complex was a substrate of IPMK, but not of classical PI3K[16]. When SF1 levels were knocked down with shRNA (Fig. 4i), the magnitudes of PLCδ-PH and Btk-PH domain accumulations decreased significantly (Fig. 4j) and were rescued by overexpression of shRNA-resistant Myc-DDK-tagged SF1 wild type (shResA-SF1-WT-Myc-DDK). This result implied that there was recruitment of the SF1-$PIP_2$ complex at the damage site and SF1-EGFP accumulated at damage sites although with a longer time constant (Fig. 4l). Since the accumulation of Btk-PH domains at damage sites depended upon IPMK phosphorylation or SF1-$PIP_2$ complexes, it was logical to postulate that knock down of IPMK would inhibit ATR recruitment.

**IPMK depletion suppresses ATR recruitment and activation.** IPMK was depleted in U2OS cells to test its role in PIKKs recruitment. In this experiment, the control and IPMK-depleted U2OS cells were separately seeded on grid-patterned dishes, irradiated, and then incubated for 1–10 min before fixation. Cells were then stained with anti-ATR and anti-γH2AX antibodies. With control siRNA, endogenous ATR was recruited to the damage sites in a time-dependent manner and co-localized with the staining of γH2AX (Fig. 4m) but was dramatically reduced with IPMK-depletion (Fig. 4n), while the expression of ATR was unaffected (Supplementary Fig. 4e). The time scale for endogenous ATR recruitment to the damage site by immunofluorescence was similar to that measured with GFP-ATR-expressing cells (Fig. 2b–f). The repair of DNA damage inferred from the decrease in γH2AX signal was slower in IPMK-depleted cells. The link between IPMK, SF1 and ATR-mediated signaling was strengthened by showing that Chk1 phosphorylation was suppressed in both IPMK- and SF1-depleted cells upon UV damage (Fig. 4p, q). Thus, these results indicated that ATR recruitment was dependent on IPMK and was likely mediated by the kinase activity of IPMK to phosphorylate the SF1-$PIP_2$ complex.

The same experiments were performed with the immunostaining of ATM and DNA-PKcs. Unlike ATR, the recruitment of endogenous ATM and the accumulation of γH2AX foci increased significantly in the absence of IPMK (Supplementary Fig. 4a, b), while the recruitment of endogenous DNA-PKcs was unaffected (Supplementary Fig. 4c, d). The levels of protein recruitment for all three PIKKs were quantified in Fig. 4o. This indicated that the mechanisms for ATM and DNA-PKcs recruitment were different from that of ATR and independent of nuclear PPIs. The fact that the recruitment of ATM was enhanced in the absence of IPMK was likely to compensate for the impaired damage response due to the absence of ATR.

**Early ATR recruitment is inhibited by Lat A and wortmannin.** The next question was how nuclear PPIs mediated the recruitment of ATR and ATRIP in an RPA-independent manner. One logical candidate was nuclear actin filament assembly. Earlier studies indicated that DNA damage promoted nuclear actin filament assembly for DNA damage clearance[33,34]. Many double-strand break (DSB) repair proteins were associated with filamentous actin in an F-actin pull-down assay, and latrunculin A (Lat A) treatment altered binding as well as chromatin association[33]. In fact, Lat A increased cellular sensitivity to ionizing radiation and reduced DNA repair capacity[45].

To test for actin involvement in ATR recruitment, Lat A was added before microirradiation and it inhibited the recruitment of GFP-ATR relative to control cells (Fig. 5a, b) in a dose-dependent manner (Fig. 5d, e). Further, recruitment of endogenous ATR was suppressed by 30% with 0.8 μM Lat A (Fig. 5h); whereas endogenous ATM accumulation was not altered and DNA-PKcs accumulation increased by 30% (Fig. 5i, j). The fact that Lat A suppressed Chk1 phosphorylation after global UV irradiation (Fig. 5k) showed generality beyond laser microirradiation. Noticeably, Lat A caused increased phosphorylation of histone H2AX even without UV exposure, consistent with earlier findings[46]. Thus, the recruitment of ATR depended upon polymerized actin.

To further prove that PI3K activity of IPMK was required in mediating ATR signaling, wortmannin was added and found to inhibit ATR recruitment in a dosage-dependent manner (Fig. 5c, f, g). Thus, recruitment of ATR depended upon both PI3K activity and actin polymerization implying that nuclear $PIP_3$ mediated actin polymerization at damage sites was likely required for ATR recruitment.

**Fig. 4** IPMK and SF1 depletion suppresses PH domain accumulation and IPMK is required for ATR recruitment. **a–d** Western blots showing effective knock-down or knock-out with or without rescue of selected phosphoinositide kinases: PIP5K1A (siRNA), PIP5K1B (shRNA), PIK3CB (CRISPR-Cas9) and IPMK (siRNA) together with **e–h** the quantified accumulation magnitude of Btk-PH and PLCδ-PH after background subtraction in designated protein depleted backgrounds ($n \geq 15$ for all cases). In panel d&h, the cells were rescued with either siRNA-resistant and flag-tagged wild type human IPMK (siRes2-3xFlag-hIPMK-WT) or its kinase dead mutant (siRes2-3xFlag-hIPMK-D144N). Numbers labeled next to the blots indicated the molecular weight of the ladder. **i** Western blot showing the effective depletion and rescue of SF1 by shRNA and an shRNA-resistant Myc-DDK-tagged mouse SF1 construct (shResA-mSF1-WT-Myc-DDK). The overpression of SF1 rescue gave two separate bands, likely with and without the Myc-DDK tag. **j** The quantified accumulation magnitude of Btk-PH and PLCδ-PH with background subtraction in SF1 depleted and rescued conditions. **k** Schematic depicting the SF1-$PIP_2$ complex as a substrate of IPMK but not classical PI3K while both kinases phosphorylate $PIP_2$ in vitro. **l** Confocal montage of transiently expressed SF1-EGFP upon laser microirradiation. **m**, **n** ATR and γH2AX immunofluorescence micrographs of U2OS nucleus upon laser microirradiation over time for control and IPMK-depleted conditions, respectively. **o** Normalized quantification comparing the magnitude of recruitment of ATR ($n = 20$), ATM ($n = 13$), DNA-PKcs ($n = 14$) and γH2AX ($n = 47$) within ROIs in control and IPMK-depleted conditions. Please refer to Supplementary Fig. 4 for the representative images of ATM and DNA-PKcs staining. **p** Transient depletion of IPMK in U2OS cells and the **q** transient depletion of SF1 in MEF cells both demonstrated a suppressed phosphorylation of Chk1 upon global UV irradiation. Data are representative of three independent experiments. in all panels. Dashed circles indicate the ROI for laser microirradiation. Error bars represent mean ± s.e.m in all panels. Student's t-test, **p < 0.01; ***p < 0.001. n.s. represents not significant. Scale bars are 10 μm for all panels

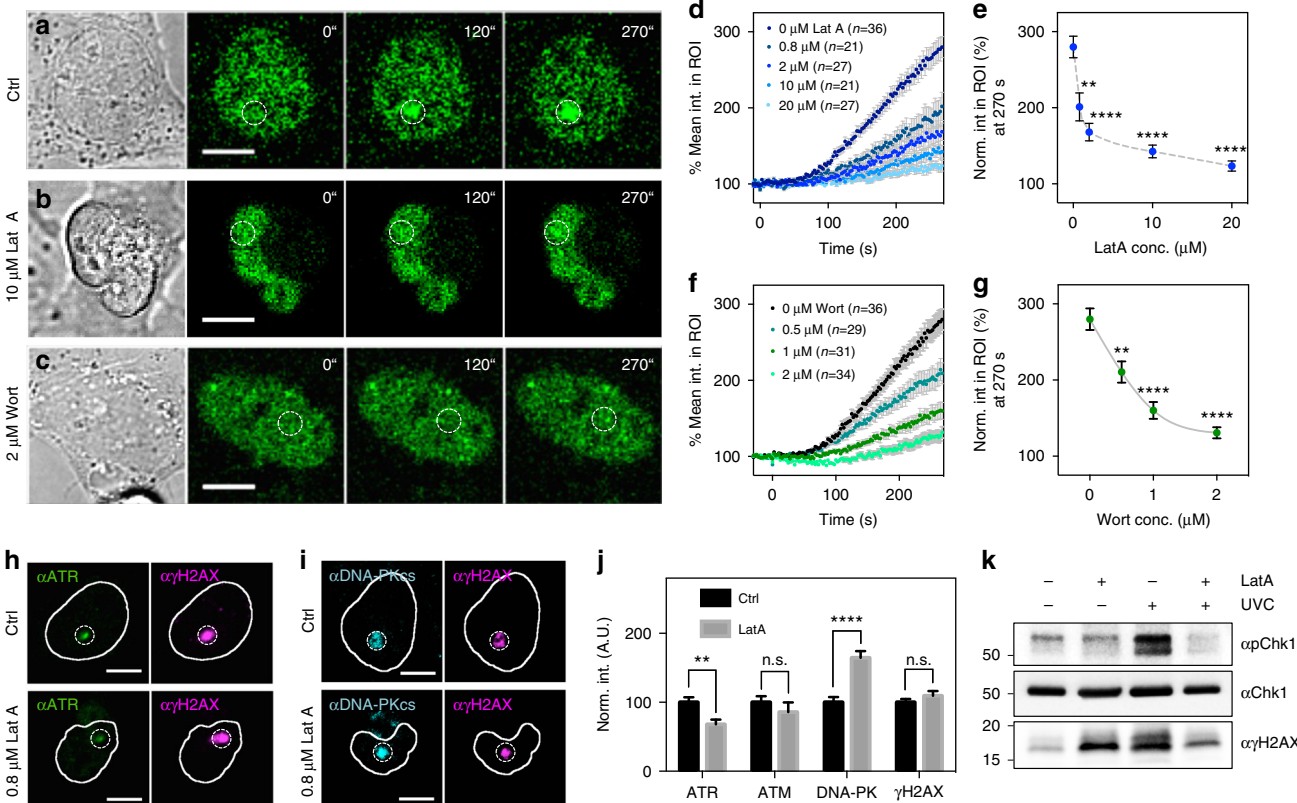

Fig. 5 Recruitment of ATR but not ATM or DNA-PKcs depends on filamentous actin assembly. Confocal montage together with the bright field images of cells stably expressing GFP-ATR showing the recruitment of ATR to damage sites **a** control conditions, **b** 10 μM Lat A and **c** 2 μM wortmannin. **d** GFP-ATR recruitment at damage sites was suppressed by Lat A in a dose-dependent manner. **e** Corresponding quantification of accumulation magnitude of GFP-ATR at 270 s after microirradiation dropped with increasing Lat A concentration from 0.8 to 20 μM. **f** The recruitment of GFP-ATR was also inhibited by wortmannin in a dose-dependent manner. **g** Corresponding accumulation magnitude of GFP-ATR at 270 s after microirradiation dropped with increasing wortmannin concentration from 0.5 to 2 μM. Representative immunofluorescent micrographs stained for **h** ATR/γH2AX and **i** DNA-PKcs/γH2AX after microirradiation in the presence or absence of 0.8 μM Lat A. **j** Corresponding normalized quantification of panel h&i and the staining for ATM (image not shown) ($n = 13$ for ATR; $n = 15$ for both ATM and DNA-PK; $n = 43$ for γH2AX staining). **k** Pre-treatment of Lat A at 10 μM for 30 min significantly suppressed the activation of Chk1 by ATR upon global UV irradiation. Error bars represent mean ± s.e.m. in all panels. Dashed circles indicate the ROI for laser microirradiation. Scale bars are 10 μm for all panels. Data are representative of three independent experiments. Student's $t$-test, **$p < 0.01$; ***$p < 0.001$; ****$p < 0.0001$. n.s. represents not significant

**Assembly of nuclear actin following PH domain accumulation**. Next, we examined nuclear actin assembly at damage sites and dependence on nuclear PIP2. Nuclear actin filaments were visualized with the nuclear F-actin marker, Utr230-EGFP-3xNLS[47]. Utr230-EGFP accumulated at damage sites, but accumulation was suppressed below background level by pre-treating cells with Lat A (Fig. 6a, b). More importantly, the accumulation of Utr230-EGFP was abolished by the co-expression of mCherry-fusion 3xNLS-PLCδPH (Fig. 6a, b), indicating that the assembly of nuclear actin at the damage site was driven by concentrated nuclear PPIs. Since PPIs were implicated in cytoplasmic actin filament polymerization by regulating the localization of formins such as mDia1 and mDia2, it was logical to test if nuclear PPIs promoted damage-induced nuclear actin assembly and whether nuclear actin polymerization was catalyzed by recruiting formins to damage sites.

The relative dynamics of the recruitment of PPIs, formins, and actin filaments to the damage sites indicated that nuclear formins and nuclear actin were downstream effectors of nuclear PPIs. As hypothesized, halftimes for recruiting either NLS-tagged mDia2 or Utr230 at damage sites were longer than that of NLS-PH domains (Fig. 6c, e). While the curve for PLCδPH was fitted properly with a single exponential rise, the curves of mDia2 and

Utr230 recovery followed a double exponential rise with the fast component leveling off at 100% fluorescence intensity. The halftimes of the slower components corresponded to halftimes of 2 and 4 s, respectively. Fluorescence recovery after photobleaching (FRAP) measurements of cells expressing NLS-tagged PLCδPH, mDia2 and Utr230 showed that all three components were equally diffusive in the nucleus (Fig. 6d). Thus, local accumulation of PIP2 served as an early DDR element that mediated nuclear actin assembly possibly by formin, and was required for ATR-dependent damage signaling, but not ATM or DNA-PKcs as illustrated in Fig. 7.

**Discussion**

Our findings indicate that PPI levels rapidly increase in response to DNA damage and are important components for ATR recruitment. Sequestration of nuclear PPIs by over-expressing PH domains with 3xNLS tags blocks the recruitment of ATR to DNA damage sites whereas mutations in the lipid-binding sites do not affect ATR recruitment. The PH domains are rapidly recruited to the damage sites with a halftime about 0.8 s, significantly faster than the recruitment of actin to those sites. Further, nuclear PPI sequestration and the addition of LatA blocks the accumulation

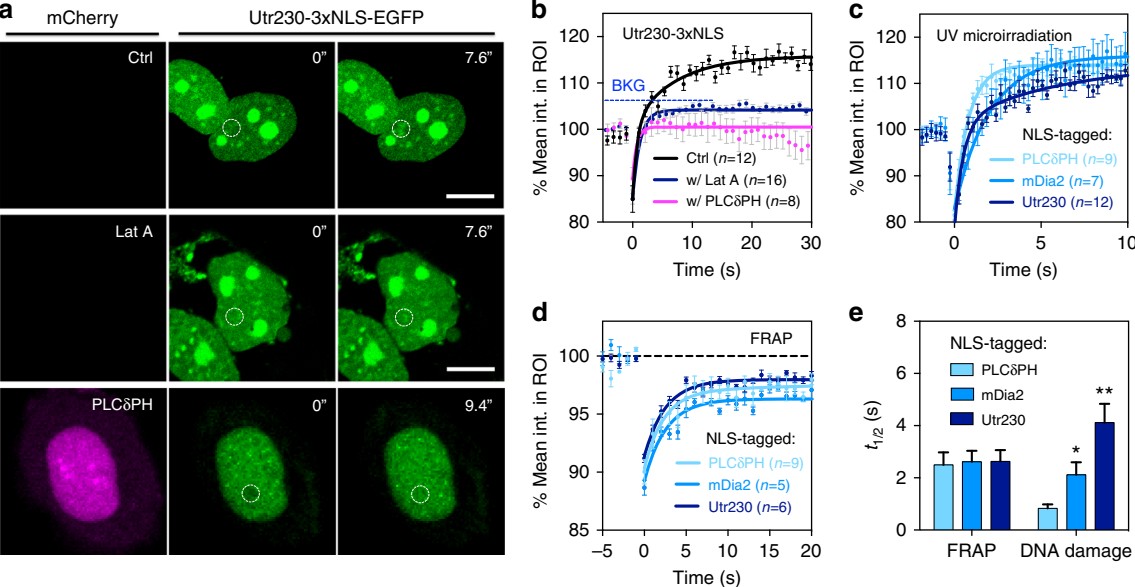

**Fig. 6** Nuclear actin assembly follows the accumulation of PH domains and formins at damage sites and is inhibited by Lat A. **a** Confocal micrographs of U2OS cells expressing NLS-tagged Utr230-EGFP before and after laser microirradiation with or without 0.8 μM Lat A or after the co-expression of 3xNLS-PLCδ-PH-mCherry. Dashed circles indicate the ROI for laser microirradiation **b** Corresponding quantification of mean fluorescence intensity of Utr230 within the ROI in the absence or presence of Lat A, or the co-expression of PLCδ-PH in the nucleus. **c** The recruiting dynamics of NLS-tagged PLCδ-PH, mDia2 and Utr230 upon laser microirradiation in independent measurements. **d** Complementary FRAP measurements of the same three NLS-tagged proteins showed that all three proteins are equally diffusive in the nucleus. **e** Summarized half times for damage-induced accumulation and photobleaching recovery times of NLS-tagged PLCδ-PH, mDia2 and Utr230. Data are representative of three independent experiments for panel A&B and at least two independent experiments for panel C-E. Error bars represent mean ± s.e.m. in all panels. Student's t-test, *p < 0.05; **p < 0.01; ***p < 0.001. n. s. represents not significant. BKG represents background level

of both nuclear actin and ATR. Of the many different PPI kinases, knock down of IPMK has the greatest effect on PPI synthesis, ATR recruitment and Chk1 activation in response to DNA damage. IPMK is sensitive to wortmannin that blocks ATR recruitment. Nuclear PPI sequestration however does not affect the recruitment of other damage repair proteins including the MRN and Ku70-80 complexes, ATM and DNA-PKcs.

Among many damage repair proteins, the MRN and Ku70-80 complexes are generally considered to be rapidly recruited to the damage sites. Their recruitment happens on a time scale from merely a second to tens of seconds[48]. In contrast, we find that the accumulation of nuclear PPIs upon laser microirradiation happens at a sub-second time scale, using EGFP-tagged, PPI-binding domains as reporters.

PPI sequestration has a direct impact on ATR recruitment, which mediates the response to both DNA double-strand breaks and stalled replication forks. The fact that nuclear PPI sequestration suppresses Chk1 phosphorylation at the damage site, but does not have an impact on RPA recruitment and phosphorylation indicates that PPI-dependent ATR recruitment in this experimental setting might follow a non-canonical signaling pathway of ATR. This observation could relate to a recently discovered ATR signaling response induced by nuclear mechanical stress that is independent of RPA[49]. Alternatively, it is possible that the PPI-dependent process leading to ATR recruitment and Chk1 phosphorylation represents an initiation event and the establishment of RPA-ssDNA nucleofilaments contributes to the amplification of the ATR signaling cascade.

By enabling PPI-mediated recruitment of ATR, IPMK appears to play a central role in this signaling axis. IPMK indirectly mediates ATR recruitment by regulating the magnitude of the PIP$_3$-binding PH domain accumulation through phosphorylation of the SF1-PIP$_2$ complex. The depletion of IPMK and SF1 results in genome

instability[40,50]. In addition, IPMK also regulates DNA damage repair through slower processes involving reduced PIP$_3$ production and accumulation of nuclear PIP$_2$ speckles, correlated with the down-regulation of the export of mRNA encoding homologous recombination-related genes[40]. As a result, IPMK-deficient cells have lower levels of homologous recombination repair proteins, but not of other repair factors involved in NHEJ that are usually mediated by ATM[40]. Thus, IPMK knockdown has long term effects on DNA damage repair that are separable from the current findings on IPMK's role in early DNA damage response. To minimize this complication, all of our laser microirradiation experiments are within 6–12 h after PH domain transfection. We assume there is no significant change in the protein expression profile within this period, especially for homologous recombination repair proteins. Thus, there are several functions that could involve IPMK but the redundant nature of the DNA damage responses makes it difficult to analyze them separately.

Recent evidence indicates that inositol metabolites also promote DNA repair[51]. Moreover, mutants in the *S. cerevisiae* gene, *ARG82*, encoding the yeast IPMK, exhibit DNA damage sensitivity[51] and have genetic interactions with mutants that are altered in the DNA damage response pathway mediated by Mec1$^{ATR}$ and its downstream regulatory kinases Rad53 and Dun1[52,53]. This indicates that the functional interactions between IPMK and the ATR response are conserved throughout evolution.

Although we find that the depletion of catalytic subunits of class I PI3Kβ, p110β, does not regulate the magnitude of PH domain accumulation, p110β is involved in DDR through a different mechanism. Intriguingly, p110β accumulates at the damage site upon laser microirradiation with a half time around 60 sec, and its kinase-independent interaction with NBS1 is critical for its recruitment[27]. It should be noted that the half times of those results are not directly comparable to ours since the laser

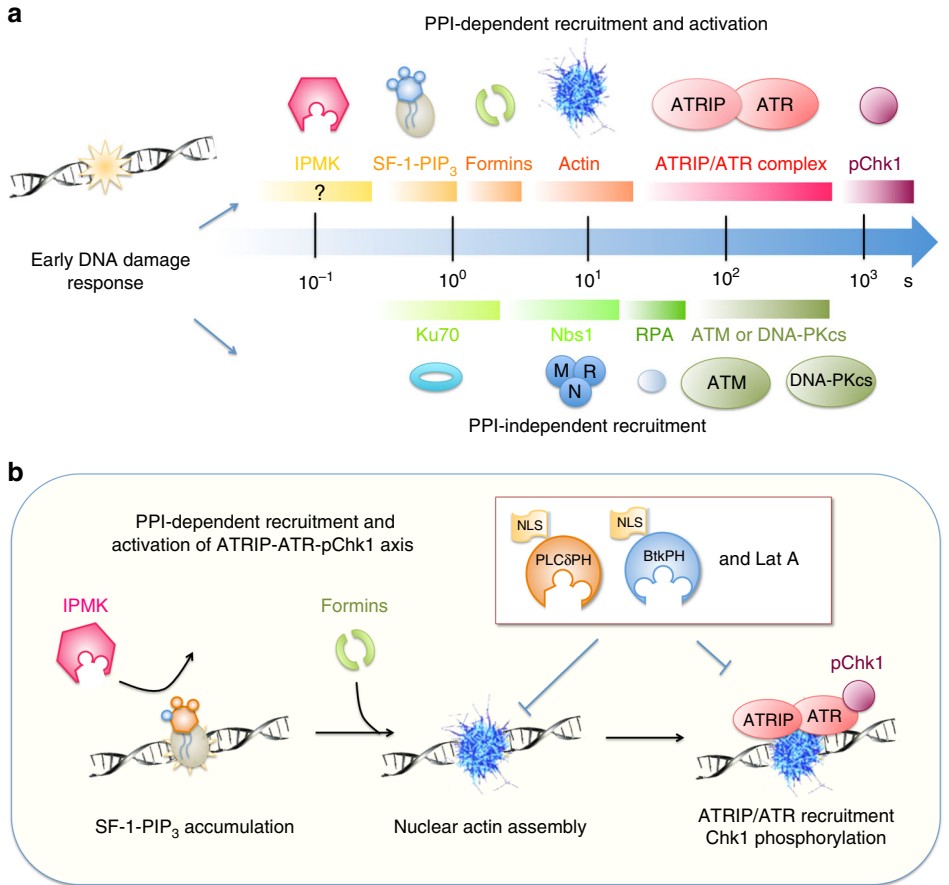

**Fig. 7** Schematic illustration of proposed model for PPI- and nuclear actin-dependent ATR-ATRIP-pChk1 signaling axis. **a** Proteins that are recruited to the damage site along the time line are grouped into two categories by whether or not their recruitment or activation depends upon nuclear PPIs. **b** Schematic presentation of proposed model for PPIs- and actin-dependent recruitment and activation of ATR-ATRIP complex

microirradiation systems are different[54]. Therefore, it appears that both p110β and IPMK can help to maintain the genome integrity.

Results presented in this study and many different recent findings support the hypothesis that ATR is recruited to the damage site through PPI-mediated local actin polymerization. First, ATR recruitment is inhibited by Lat A treatment (Fig. 5a, b), and Lat A suppresses the accumulation of Utr230-EGFP-3xNLS at DNA damage sites in the nucleus. Mass spectrometry analysis indicates that there is an interaction between ATR and factors involved in actin metabolism. Also, anti-ATR immuno-gold labeling decorates ATR on actin filaments (Kidiyoor and Foiani, personal communication). This is consistent with a recent finding that actin cytoskeleton and actin-binding proteins often contain clusters of S/TQ motifs in a SCD domain, and SCD domain proteins interact with ATR or ATM[55]. A recent study by Mullins et al. indicates that DNA damage induces the assembly of nuclear actin that is required for efficient clearance of double-strand breaks[34]. As we show in this study, NLS-tagged mDia2 and nuclear actin are sequentially recruited to the damage site within the first few seconds following the accumulation of nuclear PPIs. In addition, the co-expression of the PPI binding PH domains abolishes the accumulation of nuclear actin, indicating that nuclear actin polymerization depends on PPIs in the nucleus. Together, these observations support a model with f-actin recruitment of ATR by a direct or indirect interaction (Fig. 7).

The mechanism by which IPMK is activated for SF1-PIP$_3$ production remains elusive. A hint is provided by our results with cells pre-treated with a histone deacetylase (HDAC) inhibitor,

Trichostatin A (TSA). TSA pre-treatment nearly abolishes the recruitment of both PPI-binding domains and SF1-EGFP at damage sites (Supplementary Fig. 5a, b). TSA sensitivity is partially explained by the fact that SF1 is regulated by a well-characterized histone acetyltransferase (HAT), p300. The HAT p300 acetylates SF1 at its KQQKK motif (a.a.106–110) and regulates its DNA binding capability[56]. However, how SF1 senses damaged DNA is still an open question. TSA also causes decondensation of the heterochromatin, an increase in the nuclear volume and therefore a decrease in the intrachromosomal tension. Thus, another possible explanation involves mechanically driven accumulation of the SF1-PIP$_2$ complex as a result of the DNA lesions. Intriguingly, chromatin remodeling protein complexes, FACT (FAcilitates Chromatin Transcription) also possess PH domains, implying that PIP$_2$ might also play a role in modulating chromatin conformation[57].

In conclusion, we establish a direct link between nuclear PPIs and ATR-dependent responses to localized DNA damage. The fact that PH domains are recruited at early time points suggests that nuclear PPIs serve as a very upstream signal in mediating damage repair protein recruitment. Further, IPMK and the SF1-PIP$_3$ complex are major players along this signaling axis. Our findings provide an understanding of an important PPI function in the nucleus and indicate that the SF1-PIP$_3$ complex serves as a signaling interface in DNA damage responses.

## Methods

**Cell culture**. Mouse embryonic fibroblasts (MEF, RPTPα$^{+/+}$ strain) were obtained by immortalizing primary MEF cells isolated from RPTPα$^{+/+}$ mouse embryos at

E13-15[58], and were characterized previously by our group[59]. Human osteosarcoma U2OS cells were obtained from the American Type Culture Collection (catalogue no. HTB-96[TM]) and were used to derive lines stably expressing GFP-ATR by repeated neomycin selection following transient transfection using Lipofectamine 2000 (ThermoFisher). All three cell lines were maintained in high glucose (4.5 g L[−1]) Dulbecco's Eagle Medium (DMEM, ThermoFisher Scientific) supplemented with 10% fetal bovine serum (FBS, ThermoFisher Scientific) without antibiotics at 37 °C in 5% $CO_2$. Both cell lines were tested for mycoplasma contamination using the MycoAlert[TM] PLUS mycoplasma detection kit (Lonza) (LT07-701).

**Plasmid constructs and DNA manipulations.** Plasmids encoding phosphoinositide-binding domains, PLCδ-PH-EGFP, Btk-PH-EGFP, EGFP-P4M-SidMx2 and mCherry-P4M-SidM were a gift from Dr. Tamas Balla (Addgene)[60]. Triple repeats of an NLS tag (DPKKKRKV) derived from SV40 large T-antigen were introduced into the above-mentioned constructs at the 5′-end of the open reading frame by adding restriction sites around the NLS tag through PCR. The NLS tag was inserted in the XhoI-EcoRI sites for both Btk-PH-EGFP and PLCδ-PH-EGFP, while it was inserted in the NheI-AgeI sites for EGFP-P4M-SidMx2. Site-specific mutants of the PPI-binding domains: PLCδ-PH-R40L and PLCδ-PH-K30L/K32L/R40L[60–62], Btk-PH-R28C[63,64], and SidM-P4M-K23A[65] were generated either by a QuickChange II XL kit (Stratagene) or a Q5® Site-directed mutagenesis kit (New England Biolabs) following manufacturer's instruction. Three 3xNLS-tagged constructs: PLCδ-PH, Btk-PH and SidM-P4M together with their mutants were further subcloned into a pmCherry-N1 vector for multicolor imaging purposes. EGFP-3xNLS, EGFP-Utr230-3xNLS and EGFP-RPEL1-3xNLS were a gift from Dr. Dyche Mullins (Addgene)[47]. GFP-fusion and NLS-tagged constitutively active mDia2, GFP-NLS-mDia2 (411–1171), was a gift from Dr. Shuh Narumiya[66]. This construct was made constitutively active by deleting its N-terminal GTPase-binding domain (GBD) and disrupting the autoinhibitory interaction with the diaphanous autoregulatory domain (DAD) at the C-terminal. EGFP-FLAG-Ku70 and EGFP-FLAG-Ku80 were a gift from Dr. Steven Jackson (Addgene)[67]. TurboGFP-fusions of IPMK, NBS1, SF1 and PHLDA3 and Myc-DDK-tagged mouse SF1 were obtained from OriGene (Rockville, MD, USA). The turboGFP tags of these constructs were replaced with an EGFP tag at the C-terminal by conventional molecular biology techniques. GFP-ATR was from Dr. Randal Tibbetts[49,68]. Sequences of all cloning and sequencing primers used were listed in Supplementary Table 1.

Small hairpin RNAs (shRNA) in a pRFP-C-RS vector targeting mouse PIP5K1B and mouse SF1 were obtained from OriGene with targeting sequences: 5′-TTGAAGCTGCTTCCTTAGCGAC-CACAATA-3′ and 5′-CTTCTCTAACCGCACCATCAAGTCTGAGT-3′, respectively. Myc-DDK-tagged SF1 constructs that are resistant to this specific shRNA (shResA-mSF1-WT-Myc-DDK) were made by replacing the shRNA-targeting sequence of Myc-DDK-tagged mouse SF1 construct with the following silently mutated sequence: 5′- ATTTTCGAATCGAACAATAA AGTCGGAAT-3′ using the Q5® Site-directed mutagenesis kit (New England Biolabs). CRISPR-Cas9 KO and homology-directed repair (HDR) plasmids to knock out PIK3CB (also known as p110β) were obtained from Santa Cruz Biotechnology (TX, USA). ON-TARGETplus SMARTpool siRNA targeting human PIP5K1A was obtained from GE Dharmacon (CO, USA) with four targeting sequences: (1) 5′-ACACAGUACUCAGUUGAUA-3′; (2) 5′-GCACAACGAGAG CCCUUAA-3′; (3) 5′-GUGGUUCCCUAUUCUAUGU-3′; (4) 5′-GUAAGACCC UGCAGCGUGA-3′. Small interfering RNA (siRNA) targeting human IPMK with targeting sequence: 5′-CAUAAUGGGUACUGCUUAAUU-3′ was also obtained from GE Dharmacon. Flag-tagged human IPMK constructs together with kinase dead mutants that are resistant to this siRNA (siRes-3xFlag-hIPMK-WT, siRes-3xFlag-hIPMK-D144N) were a generous gift from Dr. Ashok Venkitaraman. The siRNA-resistant constructs were made by replacing the siRNA targeting sequence with a silently mutated sequence: 5′-CACAATGGCTACTGTTTAA-3′. All constructs were sequenced through service provided by Axil Scientific Pte Ltd. after being amplified with a Plasmid Midi Kit (Qiagen).

**Transient protein expression.** All EGFP- and mCherry-fusion constructs used in this study, except GFP-ATR, were transiently expressed in MEFs or U2OS cells using Neon electroporation transfection system (Invitrogen). The expression of NLS-tagged PPI-binding domains, especially PLCδ-PH, caused blebbing in many cells. While nuclear PPIs also regulate ALY-mediated mRNA export[39,40], only cells transfected with NLS-tagged PPI-binding domains within 6–12 h were used for laser microirradiation experiments. Besides PPI-binding domains, overexpression of constitutive active mDia2, GFP-NLS-mDia2 (411–1171), in selected cell lines caused spindle-shaped nuclei and large aggregations in the nucleus 4–6 h after transfection and it put cells in a quiescent state. Therefore, experiments for GFP-NLS-mDia2 (411–1171)-expressing cells were performed all within 2–4 h after transfection. In contrast, the expression of nuclear F-actin marker, EGFP-3xNLS-Utr230, showed no obvious adverse effect on cells and experiments were performed within a wider time range from 4–24 h.

**Laser microirradiation and diffusivity measurements.** Laser microirradiation for localized DNA damage induction was performed using a UV laser (355 nm; 1 kHz

repetition rate; PowerChip, Teem Photonics) with a 60X objective (NA 1.4, Nikon). The UV laser was introduced into the microscope through a custom-made UV dichroic mirror with a mechanical shutter controlled by a homemade ImageJ plugin. The power of UV laser used in this study ranged from 0.5 to 15 nW measured at the back aperture of the objective. For convenience, the power indicated in this study refers to the power measured at the back aperture of the objective and was set at 1 nW for most experiments if not further indicated. To capture the sub-second accumulation of PH domains and a few other proteins, high time resolution imaging was performed using Nikon A1R confocal microscope (Nikon) with a resonance scanner, which gave an average frame rate of 3.7 per second, 270 ms per frame. Imaging with Galvano scanner was selected for studying slower dynamics such as the accumulation of GFP-ATR, with a frame rate at about 0.5–1 per second, 1–2 s per frame. Cells expressing constructs of interest were stained with 10 µg mL[−1] Hoechst 33342 (Sigma-Aldrich) for 10 min, which served as a photosensitizer for DNA damage induction, prior to DNA damage experiments. Localized DNA damage was introduced by exposing the nucleus to a 355 nm (UVA) laser for 500 milliseconds at 1 nW laser power, at the back aperture of the objective. To compare the diffusivity of each probe, fluorescence recovery after photobleaching (FRAP) experiments were performed. Circular ROIs of 2 µm in diameter were selected in the nuclear matrix outside of nucleoli. The photobleaching time was limited at 0.25 s to avoid significant diffusive recovery and recovery halftimes were determined using standard techniques following a double normalization method.

**Global DNA damage induction.** UV-induced DNA damage was achieved by sensitizing the cells with Hoechst 33342 (Sigma-Aldrich) at 10 µg mL[−1] for 10 min and followed by being exposed to a UVP 3UV Ultraviolet Lamp (Thermo Scientific) at 254 nm at 1.33 W m[−2] for 2 or 10 min as indicated. The power delivered onto a sample surface was measured by a HS116K silicon photodiode (BaseLine Chromtech, China) with a 1.5 cm spacer placed between the lamp and the sample. Chemical-induced damage, on the other hand, was caused by treating cells with 0.01% MMS (Sigma) for 2 hrs. In both cases, damaged cells were fixed either right after damage induction or were allowed to recover for an hour in an incubator before fixation, and the cells were subjected to immunostaining. For UV treated cell sample preparation for western blotting, cells were seeded in 10 cm dishes and sensitized with Hoechst33342 at 10 µg mL[−1] for 10 min at one hour prior to exposure to UV. Drug treatment such as Lat A was added 30 min prior to UV treatment. The cell-containing 10 cm dishes were filled with minimum 5 mL medium during UV treatment. Cells were placed under UV lamp at two perpendicular orientations for 1 min each to ensure homogenous UV exposure. All UV-treated cells were allowed to recover for an hour in an incubator before harvesting.

**Immunofluorescence.** DNA damage was confirmed by immunostaining the cells with an anti-phospho-γH2AX Ser139 antibody (Abcam). After DNA damage induction either by UV, by chemical, or by microirradiation, MEF and U2OS cells were fixed in 4% formaldehyde (Pierce) at room temperature (RT) for 20 min in phosphate-buffered saline (PBS). The samples were then washed, permeabilized with 0.3% Triton-X 100 in PBS for 10 min, quickly washed with PBS and then blocked with 3% bovine serum albumin (Sigma-Aldrich) in 0.1% Triton-X 100 in PBS for 30 min. Primary antibodies such as rabbit anti-ATR monoclonal antibody (Cell Signaling Technologies) and mouse anti-phospho-histone H2A.X Ser139 (Merck Millipore) were sequentially incubated at room temperature for an hour each. Plates were quickly washed with blocking buffer before incubating with blocking buffer containing both of the secondary antibodies, AlexaFluor 546 anti-rabbit IgG and AlexaFluor 647 anti-mouse IgG (Invitrogen) for 30 min. The plate was washed three times after secondary antibody staining and stained with Hoechst 33342 (Sigma-Aldrich) at 10 µg mL[−1] for 10 min during the final wash and stored in PBS for imaging. The following are the primary antibodies and conditions used in immunofluorescence assays: mouse anti-PIP₂ [clone 2C11] (ab11039, 1:500) and rabbit anti-phospho-histone H2A.X Ser139 (ab2893, 1:1000) were from Abcam. Mouse anti-phospho-histone H2A.X Ser139 [clone JBW301] (05–636, 1:1000) was from Merck Millipore. Rabbit anti-ATR [clone E1S3S] (13934, 1:500), rabbit anti-ATM [clone D2E2] (2873, 1:500) and mouse anti-DNA-PKcs [clone 3H6] (12311, 1:500) were from Cell Signaling Technology. Rabbit anti-Phospho RPA32 Ser4/Ser8 (A300-245A) (1:500), rabbit anti-Phospho RPA32 Ser33 (A300-246A, 1:500) and rabbit anti-RPA32 (A300-244A, 1:500) were from Bethyl Laboratories. Rabbit anti-Phospho Chk1 Ser345 [clone 133D3] (2348, 1:500) was from Cell Signaling Technology. Corresponding secondary antibodies include: AlexaFluor546-donkey anti-rabbit IgG (10040, 1:1000), AlexaFluor546-goat anti-mouse IgG (A11003, 1:1000), AlexaFluor568-goat anti-rabbit IgG (A21069, 1:1000), AlexaFluor647-goat anti-mouse IgG (A21237, 1:1000) and AlexaFluor647-donkey anti-rabbit IgG (A31573, 1:1000). Samples were imaged with Nikon A1R confocal microscope (Nikon) using a Galvano scanner and a pinhole size at 40 µm. Single optical slices with highest fluorescent intensity in the ROI were selected for representative images.

**RNA interference and western blotting.** Knocking down in MEF is achieved by transfecting cells with shRNA at 0.1 µg µL[−1] using Neon electroporation transfection system with a 10 µL pipette kit (Invitrogen) for laser microirradiation or

100 µL pipette kit (Invitrogen) for western blots sample preparation. Puromycin selection was used at 8 µg mL$^{-1}$ for 48 hrs when the knock down efficiency is low. Knock down in U2OS was achieved by transfecting cells with siRNA using a Lipofectamine® RNAiMAX reagent (Thermo Fisher Scientific) following manufacturer's instruction. Cells were harvested 72 hr later, pelleted and washed with PBS before being lysed in a RIPA lysis buffer (Sigma-Aldrich) supplemented with 1X cOmplete protease inhibitor cocktail (Roche). Concentrated samples were run on a 4–20% Mini-PROTEAN® TGX™ (Bio-Rad) precast protein gels and transferred onto membranes. Membranes were blocked with 5% Bovine Serum Albumin (BSA) in 1X TBST (Tris-Buffered Saline with Tween-20) for 1 h, and incubated with primary antibody overnight at 4 °C. The membranes were washed three times in TBST (10 min per wash), secondary horseradish peroxidase (HRP) antibodies were added at 1:2000 for mouse or rabbit IgG (Bio-Rad) for one hour at room temperature in 1X TBST, and the membranes were washed again three times in TBST. The chemiluminescence of the membranes was developed using a Super Signal West Femto Substrate Kit (Pierce) followed by exposing the membranes to a CL-Xposure film (Thermo Scientific Pierce) in a dark room or to a ChemiDoc Touch Imager (Bio-Rad). For loading control analysis, blots were stripped and re-probed with mouse anti-α-tubulin (Sigma-Aldrich). Primary antibodies and conditions used in western blotting are list as following: rabbit anti-PIP5K1A (TA343090, 1:1000), rabbit anti-PIP5K1B (TA331347, 1:1000), rabbit anti-IPMK (TA308405, 1:1000) were from Origene. Rabbit anti-PI3 kinase p110β [clone EBR5515(2)] (ab151549, 1:1000) was from Abcam. Mouse anti-α-tubulin [clone DM1A] (T9026, 1:3000) was from Sigma-Aldrich. Rabbit anti-Phospho-Chk1 (#2348, 1:500), mouse anti-Chk1 (#2360, 1:1000), rabbit anti-phospho-histone H2A.X Ser139 [20E3] (#9718, 1:1000), rabbit anti-SF1 [D1Z2A] (#12800, 1:1000) and rabbit anti-ATR [clone E1S3S] (#13934, 1:1000) were from Cell Signaling Technology (MA, USA). The secondary antibodies, HRP-conjugated goat anti-rabbit IgG (#170–6515) and goat anti-mouse IgG (#170–6516) were from Bio-Rad and were used at half of the primary antibody concentration. Uncropped scans of all blots are shown in Supplementary Fig. 6.

**Knock out by CRISPR genomic editing**. PIK3CB (p110β)-deficient MEF cell lines were generated by a CRISPR-Cas9 technique[43,69] using commercially available CRISPR Knockout (KO) and homology-directed repair (HDR) plasmids (sc-428980 and sc-428980-HDR, Santa Cruz Biotechnology, TX, USA) following manufacturer's instruction. In brief, a PI 3-Kinase p110β CRISPR-Cas9 KO plasmid together with a PI 3-kinase p110β HDR plasmid were co-transfected into MEF cells in a 1:1 ratio using a Neon electroporation transfection system (Invitrogen). The GFP tag on a PI 3-Kinase p110β CRISPR-Cas9 KO plasmid and the RFP tag on a PI 3-kinase p110β HDR plasmid allowed one to isolate dual-color labeled single cells 24 h after transfection using FACSAria cell sorter (BD Biosciences). Co-transfection of the HDR plasmid inserted a puromycin resistance gene into the genome enabling selection of stable knockout (KO) cells through the two 800 bp homology arms designed to specifically bind to the genomic DNA surrounding the corresponding Cas9-induced double strand DNA break site. Single cell colonies were then selected with puromycin-containing medium at 6 ug mL$^{-1}$, a concentration optimized by pre-selection titration using control MEF cells. The depletion of p110β expression was confirmed by western blot. The resulting p110β-KO lines showed decreased cell proliferation as described earlier[70].

**ATP depletion and drug treatments in laser microirradiation**. Adenosine triphosphate (ATP) depletion was done by incubating cells with 10 mM sodium azide (Sigma-Aldrich) and 6 mM 2-deoxy-D-glucose (Sigma-Aldrich) in a glucose-free Ringer's medium, composed of 150 mM NaCl, 5 mM KCl, 1 mM CaCl$_2$, 1 mM MgCl$_2$, 20 mM HEPES at pH 7.4. Cells were assayed 30–50 min after treatment to avoid side effects induced by a prolonged depletion of ATP. For PI3K inhibition, cells were first sensitized by Hoechst33342 (Sigma-Aldrich) at 10 µg mL$^{-1}$ for 10 min followed by treatment with wortmannin (Selleckchem) at 0.5–2 µM for 3 hrs prior to laser microirradiation. The procedure for latrunculin A (Lat A) treatment was similar. Lat A from Sigma-Aldrich was added following 10 min Hoechst sensitization at a concentration from 0.8 to 20 µM for at least 1 h prior to the DNA damage induction. To perform laser microirradiation on dead cells, cells were killed by the addition of 100 mM β-mercaptoethanolamine (MEA) (Sigma-Aldrich). Massive cell death happened within 30 min after MEA treatment and the laser microirradiation was performed within the next hour.

**Image quantification and data analysis**. To quantify the change in the level of PIP$_2$ staining, the mean fluorescence intensity of the whole nucleus was quantified instead of the number of speckles, since we didn't assume the distribution of PIP$_2$ in the nucleus. The mean fluorescence intensity within each nucleus was measured followed by background subtraction, averaging, then normalized against the same number measured from the non-UV treated non-transfected cells. The same procedure was adopted for the analysis of γH2AX staining so the changes in level could be compared. For analyzing protein recruitment upon laser microirradiation in live cells, the mean fluorescent intensity within a circular ROI at the damage site was quantified. A 2.5 µm diameter ROI was selected while the UV laser at 1 nW has a beam width whose full width at half maximum (FWHM) was 0.85 µm as measured using a sandwiched thin-layer fluorescently labeled agarose gel. The drifting

of ROI was negligible for most measurements wherein only the first 45 or up to 270 s after microirradiation were recorded. The average fluorescence intensity in the nearby nuclei was measured for individual images for bleaching correction purposes. The analysis was same as that for FRAP measurements which followed standardized double normalization method with a background subtraction. For quantifying protein recruitment upon laser microirradiation in fixed and stained cells, a single optical slice with maximum intensity in a circular ROI was used for intensity quantification. Same-sized ROI with 2.5 µm in diameter was used for all samples.

**Statistics**. For EGFP-tagged protein recruitment dynamics measurements, 5–10 cells were selected for each independent experiment. For immunofluorescence studies using laser microirradiation, 5 cells were microirradiated within 5 min, fixed and stained in each experiment. Experiments were repeated three times in most experiments. Data were pooled from replicated experiments for quantification. All quantification was performed within FIJI while t-test was carried out using GraphPad Prism version 6.04 (GraphPad Software, La Jolla California USA). Student's t-test was performed throughout the study for statistical significance evaluation. The notation of statistical significance was the same throughout this work:, *$p < 0.05$; **$p < 0.01$; ***$p < 0.001$; ****$p < 0.0001$; n.s. represents not significant.

**Data availability**. The data that support the findings of this study are available from the corresponding author on reasonable request.

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

## Acknowledgements

The authors thank Drs Hongying Chen, Ratna P. Vogirala, Kathirvel Paramasivam and Mr. Yunlong Liu from the Protein Cloning and Expression Core in Mechanobiology Institute for their assistance in preparing the constructs. We also thank Kin-Mun Lee and Mok Meng Huang for their technical support at the FACS sorting facility in Cancer Science Institute in National University of Singapore. The authors are grateful for the fruitful discussion from Prof. Ashok Venkitaraman. This work was supported by a grant from Singapore Ministry of Education and National Research Foundation to the Mechanobiology Institute, National University of Singapore.

## Author contributions

Y.-H.W., G.V.S., M.F. and M.S. designed the experiments. G.B. established GFP-ATR stable cell line. Y.-H.W. and A.H. carried out the experiments and Y.-H.W. performed all data analysis. Y.T. built up the laser microirradiation platform. Y.-H.W., M.F. and M.S. wrote the paper.

## Additional information

**Competing interests:** The authors declare no competing financial interests.

