## [Peer Review File · Nature Communications]

Reviewers' comments:

Reviewer #1 (Remarks to the Author):

Wang et al propose a hitherto undetected role for phosphoinositide lipids (PPI) in activation of the ATR-Chk1 branch of the DNA damage response. They propose that formation of PIP3 from PIP2 bound to the SF1 steroid receptor is catalysed at sites of DNA damage by the PI3K Inositol polyphosphate multikinase (IPMK), and that formation of PIP3 is required for the recruitment of ATR and its targeting subunit, ATRIP, via local formin-mediated actin polymerisation. This is an unexpected and original concept which, if fully substantiated, would be of considerable significance in the DNA damage signalling field.

Major points:

1) A significant weakness in the study as it stands is that almost all of the key observations have been made using laser irradiation to induce sub-nuclear DNA damage combined with immunofluorescence microscopy and FRAP to monitor the subsequent levels and distribution of PPIs, PPI-binding domains, and checkpoint signalling/ repair factors within the nucleus. Although these techniques are wholly legitimate, and very powerful in appropriate circumstances, they are also highly specialised. For that reason it is very important that some of the key experimental manipulations that are alleged to affect ATR activation should be confirmed by western blotting using the phospho-specific antibody against Chk1 pS345, the accepted gold standard marker and method for ATR activity measurements. There are a couple of reasons for this. First, although pS345 Chk1 is documented by immunofluorescence in Fig. 2J, the finding that S345-phosphorylated Chk1 accumulates at sites of damage is in itself quite surprising, as this is not a generally accepted phenomenon in the field (see for example Smits et al. *Curr. Biol.* 16: 150-9 2006). Second, in all subsequent experiments recruitment of ATR to the site of damage is considered to be equivalent to activation. While this might be the case, it seems unreasonable to make this assumption given that it is a novel, non-canonical mechanism of ATR activation that is under study. For these reasons it will be necessary to confirm that ATR activation is suppressed by western blotting using anti-pS345 Chk1 (and anti-gamma H2AX for comparison ideally) during: a) overexpression of PPI-binding domains (Fig. 2), b) depletion of IMPK and SF-1 (Fig. 4), and c) treatment with latrunculin A (Fig. 5).

2) In Supp Fig. 2G and I DNA damage-induced RPA phosphorylation at S4/ 8 and S33 is shown to be resistant to overexpression of the PIP2-binding domain. For RPA S4/ 8 this is reasonable as DNA PK has been proposed to catalyse this modification, and recruitment of DNA PK is not affected. RPA S33 by contrast is generally thought to be catalysed by ATR, so why is this modification unperturbed given that ATR is not efficiently activated under these conditions, at least as judged by recruitment to the site of damage?

3) Wortmannin is shown to oppositely affect the recruitment of the tagged Btk and PLC-delta PH domain proteins, presumably because the drug is blocking conversion of PIP2 to PIP3. Given that it is subsequently proposed that this conversion is catalysed by IPMK, obviously this implies that IPMK must be inhibited by wortmannin – is this known?

Minor points

1) Fig. 3A, B. On P8 this figure is said to show recruitment of the tagged PH domains of "TAPPI-PH", however this is not in fact shown, nor is it mentioned anywhere else in the ms.

2) P9/ 10 – wortmannin is stated to have "caused an apparent increase in PIP2 and decrease in PIP3...that was reflected in the level of PIP2 and PIP3 PH binding domain accumulation". It is appreciated that the sentence contains the qualifier "apparent", but given that PPI levels were not measured, but are being inferred from the behaviour of the PH domain reporters, it would seem

more accurate to say "as judged by the relative levels of accumulation of PIP2 and PIP3 PH binding domains".

3) P13 refers to experiments in Supp Fig. S4A, B showing that latrunculin A does not inhibit recruitment of ATM or DNA PK to sites of damage, however the figure actually contains the original western blots used in Fig. 4.

4) A related point, the original blot documenting siRNA depletion of SF-1 is technically unconvincing. Better quality data is required to secure the important point that SF-1 was in fact successfully depleted.

Reviewer #2 (Remarks to the Author):

This manuscript describes a novel role for nuclear phosphoinositide signalling in the DNA damage response. The role of nuclear PI signalling has remained enigmatic for many years, and this study shines a light on the role of PIP2 and PIP3 signalling in the nucleus and implicates IPMK in this process.

This study is well done, and definitely provides novel insights into the DNA damage response and nuclear PI signalling.

I have a few queries the authors should resolve before publication:

1. Is the recruitment of phosphoinositides to sites of DNA damage dependent on active transcription? i.e. have they repeated their experiments with transcription elongation inhibitors such as DRB or splicing inhibitors. This is an important point, given that PIP2 localises to nuclear speckle domains, which are thought to be where certain mRNAs are processed prior to their export.

2. I do have a slight concern regarding their IPMK data. I am convinced that IPMK is required for the recruitment of PIPs from their rescue data and their data on ATM compensation following IPMK depletion is nice. However, at the time points they use (66h-72h), levels of HR proteins such as RAD51 and CHK1 (and maybe ATR) will be clearly decreased due to the effect of IPMK on nuclear export of HR mRNAs, so how can they be sure that the effects on ATR recruitment aren't due to this effect? Again, their rescue data are convincing, and the fact that it has a specific effect on Btk-PH and not PLC δ -PH is important. Nevertheless, a sentence should be included discussing this alternate possibility.

3. Quantitation - as far as I can tell, there is no indication of the number of independent experimental repeats performed for each experiment - For example in Figure 4: "Normalized quantification comparing the magnitude of recruitment of ATR (n=20), ATM (n=13), DNA-PKcs (n=14) and γ H2AX (n=47) within ROIs in control and IPMK-depleted conditions. Please refer to Supplementary Figure S3 for the representative images of ATM and DNA-PKcs staining. Error bars represent mean \pm S.E.M. in all panels."

Is this from one experiment, or from multiple independent experiments?

4. They mention TSA data in the discussion but I can't seem to find the data in any figures : "The mechanism by which IPMK is activated for SF-1-PIP3 production remains elusive. A hint is provided by our results with cells pre-treated with a histone deacetylase (HDAC) inhibitor, TSA."

Reviewer #3 (Remarks to the Author):

This is an interesting article describing the accumulation of PIP(n) binding domains at sites of DNA damage in the nucleus and the impact of these domains on the ATR-dependent DNA damage response. Filamentous actin assembly is postulated to be an important factor in mediating PIP(n) driven ATRIP/ATR recruitment and Chk1 phosphorylation.

As presented, the data are convincing and controls using mutant lipid binding domains that cannot interact with their target lipid, and the lack of effect of these lipid binding domains on other, parallel, DNA damage responses go a long way to argue against non-specific effects of the manipulations used. Further, the additional effects of reducing expression of IPMK and SF1 add weight to a nuclear-specific function of phosphoinositides in the ATR damage response pathway.

Overall, this manuscript has the potential to make an important advance in our understanding of the role of nuclear phosphoinositides but, I have several specific concerns that relate mostly to the interpretation of the results.

Specific comments:

1. Have the effects of non-nuclear-localised lipid binding proteins been investigated with respect to the DNA damage response ie are we sure the effects of these domains are confined to the nucleus? Further, have the effects of the nuclear-localised domains been investigated with respect to plasma membrane signalling e.g. AKT-phosphorylation?

2. The effects of lipid binding domains selective for PI4P, PI45P2, PI34P2 and PIP3 suggest all of these lipids are involved in the ATR response. The effect of IPMK and SF1 then suggest phosphorylation of PI45P2 presented on SF1 is important.

I find the conceptual framework here difficult to understand. Are the authors saying all of these lipids are linked within some sort of nuclear membrane-localised metabolic pathway (ie PI to PI4P to PI45P2), but then PI45P2 is extracted by SF1 and phosphorylated in this state to PIP3 by IPMK - but then what can be the involvement of PI34P2, since this is thought to originate via PIP3 5-dephosphorylation or PI4P 3-phosphorylation (by Class II PI3K)?

I'm also confused by the apparent efficacy of the PH domain probes to sequester the lipids – is the BTK-PH domain imagined to bind the head group of PIP3 whilst presented on SF1? The authors need to show this is possible (BTK-PH can't be acting prior to binding to SF1 otherwise there would be no necessity for IPMK phosphorylation of PI45P2 to PIP3 on SF1)? Or is the model here that PIP3 is released from SF1 into some sort of membrane compartment before it acts to regulate actin and ATR, and it is at this point that the BTK-PH can interfere?

3. I can understand why the anti-PIP2 antibody would present a different overall image to that of the PI45P2-selective PH domain, but wouldn't you anticipate some level of co-localisation if they are both binding to the same pools of PI45P2? I couldn't discern any evidence of this.

Further, in Figure 1, wouldn't you anticipate that expression of the PI45P2-selective PH domain would reduce the number of anti-PIP2 puncta in the no-UV condition as well as 1hr after UV irradiation?

4. The manuscript would be strengthened by a clear, quantitative defect in the ATR-dependent DNA damage repair response in IPMK-KO cells.

5. The cartoon in Fig 4L is not helpful without depicting the conversion of PI45P2 to PIP3.

Reviewer #4 (Remarks to the Author):

I have enjoyed reviewing this article, the experimentation is creative and yields novel insights. In this article Wang et al report on nuclear phosphoinositide signaling following DNA damage afflicted using a UV laser (micro irradiation). The work relies on PH-domain containing peptides to recognize specific classes of phosphoinositides. These PH-domain containing peptides serve to both deplete and (because they are fluorescently tagged) quantify and localize specific phosphoinositides. Most of the work is done in a cell line where nucleolar accumulation of the PH-domain containing peptides complicates interpretation of results. However, the authors recognize this complication and select regions of interests (ROIs) outside of the nucleoli. The major finding is that there are two forms of DNA damage repair, one reliant on PIP signaling, resulting in recruitment of ATR to sites of DNA damage, the other one, resulting in recruitment of ATM and DNA-PK, is independent of PIP signaling.

The technology is sophisticated and quite indirect (the intensity of fluorescent PH-domain containing peptide is taken as a read-out for PIP concentration in a given ROI). With that comes a risk of over interpretation of the observations. The data presentation requires clarifications: Fig. 1 How is 'normalized mean intensity' measured and calculated. The segment on image quantification in the methods section only describes how ROIs re evaluated. But here the entire nucleus was evaluated. How was that done. Is green fluorescence of the entire nucleus measured, are speckles counted? By eye, the difference between the PH-domain transfected cells and the control is not different when looking at the PIP2 immunization. (A versus C). However the bar graph does show a significant difference.

Fig. 2 Convincingly shows specificity of the PH-domain containing peptides for certain PIPs, and failure to recruit ATR pathway members to sites of damage

Fig. 3 Failure of PH-domains to accumulate at sites of DNA damage in dead cells could be due to any number of reasons, they do not proof active lipid synthesis as the authors claim. (I-K) The difference in ROI fluorescence intensity is a mere 5% for the BTK-PH domain and increase for the PLC-delta PH domain likewise. While the results are statistically significant, this raises the question if they are biologically significant. Is ATR recruitment to sites of micro irradiation wortmannin-sensitive?

Fig. 4 E->J Similar issue. %Inc (percent increase I presume) is the major read-out, and while statistically significant, one wonders about it biological meaning.

Fig. 6 Don't understand subpanel D, legend too cryptic

Statistics:

Method section needs a separate statistics section that explains how many experimental repeats were done, how many cells per experiments counted, justification of those numbers, how normalization was done.

Use of the standard error of the means (SEM) rather than Standard Deviation (SD) not explained when presented such as in Fig. 4, 5 etc. Throughout the figures statistical tests need to be added and p-values presented, this is not done consistently. Also number of repeats of independent experiments (not only number of cells analyzed per experiment) need to be indicated.

Cell line: Need to explain why results were only obtained in U2OS osteosarcoma cells with the exception of MEFs in one experiment. If this mechanism is true in general, it should be shown in a range of cell lines. Is there any reason to assume that differs between normal and cancer cells?

The text needs revision of the language. It is in general too wordy, and too many acronyms are used, it is hard to follow.

The discussion fails to explain what this newly discovered mechanism might mean for normal or

cancerous cells, how it might have evolved, what the significance of the differential regulation of ATM/DNA-PK on the one and ATR recruitment on the other might be, and why the timing which the authors delineate, might be of importance.

As detailed above, the paper needs major revisions and is not acceptable for publication in its current form.

Reviewers' comments:

Reviewer #1 (Remarks to the Author):

Wang et al propose a hitherto undetected role for phosphoinositide lipids (PPI) in activation of the ATR-Chk1 branch of the DNA damage response. They propose that formation of PIP3 from PIP2 bound to the SF1 steroid receptor is catalysed at sites of DNA damage by the PI3K Inositol polyphosphate multikinase (IPMK), and that formation of PIP3 is required for the recruitment of ATR and its targeting subunit, ATRIP, via local formin-mediated actin polymerisation. This is an unexpected and original concept which, if fully substantiated, would be of considerable significance in the DNA damage signalling field.

Major points:

1) *A significant weakness in the study as it stands is that almost all of the key observations have been made using laser irradiation to induce sub-nuclear DNA damage combined with immunofluorescence microscopy and FRAP to monitor the subsequent levels and distribution of PPIs, PPI-binding domains, and checkpoint signalling/ repair factors within the nucleus. Although these techniques are wholly legitimate, and very powerful in appropriate circumstances, they are also highly specialised. For that reason it is very important that some of the key experimental manipulations that are alleged to affect ATR activation should be confirmed by western blotting using the phospho-specific antibody against Chk1 pS345, the accepted gold standard marker and method for ATR activity measurements. There are a couple of reasons for this. First, although pS345 Chk1 is documented by immunofluorescence in Fig. 2J, the finding that S345-phosphorylated Chk1 accumulates at sites of damage is in itself quite surprising, as this is not a generally accepted phenomenon in the field (see for example Smits et al. Curr. Biol. 16: 150-9 2006).*

A: Our working hypothesis regarding nuclear PIP2-mediated ATR activation has now been tested with western blots following the suggestion from this reviewer (see following points for detailed description). While our western blot results are all positive and have strengthened our hypothesis, we would like to point out that the accumulation of S345-phosphorylated Chk1 upon laser microirradiation has also been reported in a recent work of the Yokomori group in which they showed that the accumulation of pChk1 was laser power-dependent. Please refer to the

Supplementary Figure S7 in the work of Cruz et. al. Nucleic Acids Res. 2016, 44(3), e27.

2) *Second, in all subsequent experiments recruitment of ATR to the site of damage is considered to be equivalent to activation. While this might be the case, it seems unreasonable to make this assumption given that it is a novel, non-canonical mechanism of ATR activation that is under study. For these reasons it will be necessary to confirm that ATR activation is suppressed by western blotting using anti-pS345 Chk1 (and anti-gamma H2AX for comparison ideally) during: a) overexpression of PPI-binding domains (Fig. 2), b) depletion of IMPK and SF-1 (Fig. 4), and c) treatment with latrunculin A (Fig. 5).*

A: a) We tested whether NLS-tagged PH domain suppresses Chk1 activation by comparing the change of pChk1 level of cells expressing cytoplasmic EGFP, cytoplasmic PLC δ PH and NLS-tagged PLC δ PH, respectively, before and after UV irradiation. Suppressed Chk1 activation was evident in cells expressing NLS-tagged PLC δ PH domain, but not for cells expressing EGFP or cytoplasmic PLC δ PH domain (Fig. 1E). The result was quantified in Fig. 1F and the fold increase of pChk1 upon UV irradiation for EGFP vs. PLC δ PH vs. NLS-PLC δ PH-expressing cells was determined to be 1.53: 1.16: 0.37, respectively. b) The depletion of IPMK and SF1 gave similar results in which the level of phosphorylated Chk1 was about the same before being treated with UV, but was significantly lower for IPMK- or SF1-depleted cells after UV treatment (Fig. 4P&Q). c) The suppression of Chk1 phosphorylation by Lat A was also evident (Fig. 5K). Upon the treatment with LatA, the γ H2AX level of cells increased without being exposed to UV. This was consistent with the previous finding (Shin et. al., Oncology Reports 2011). The pChk1 level was about the same for control and Lat A-treated cells before being treated with UV, but was significantly lower for Lat A-treated cells, compared to control cells after UV treatment.

3) *In Supp Fig. 2G and I DNA damage-induced RPA phosphorylation at S4/ 8 and S33 is shown to be resistant to overexpression of the PIP2-binding domain. For RPA S4/ 8 this is reasonable as DNA PK has been proposed to catalyse this modification, and recruitment of DNA PK is not affected. RPA S33 by contrast is generally thought to be catalysed by ATR, so why is this modification unperturbed given that ATR is not efficiently activated under these conditions, at least as judged by recruitment to the site of damage?*

A: The Ser33 residue of RPA is a common phosphorylation site for all three PI3K-like kinases (PIKKs), as summarized in other report (Vassin et. al., JCS 2009, 122(22), 4070-80). The fact that the suppression of ATR recruitment didn't result in a decrease in RPA phosphorylation at Ser33 implies that the phosphorylation of RPA at this site might be compensated by the activation of the other two PIKKs.

4) *Wortmannin is shown to oppositely affect the recruitment of the tagged Btk and PLC-delta PH domain proteins, presumably because the drug is blocking conversion of PIP2 to PIP3. Given that it is subsequently proposed that this conversion is catalysed by IPMK, obviously this implies that IPMK must be inhibited by wortmannin – is this known?*

A: The kinase activity of IPMK has been shown to be insensitive to wortmannin *in vitro* after purification (Resnick et. al., PNAS 2005). However, the PI3K activity of IPMK was suppressed by wortmannin inside cells. Its sensitivity to wortmannin was likely conferred by p110 PI3K, directly or indirectly, as inhibiting PI3K by wortmannin prevents IPMK phosphorylation and activation (Maag et. al., PNAS 2010).

Minor points

1) *Fig. 3A, B. On P8 this figure is said to show recruitment of the tagged PH domains of "TAPPI-PH", however this is not in fact shown, nor is it mentioned anywhere else in the manuscript.*

A: We meant to compare PLC δ PH, BtkPH and the PI4P-binding P4M-SidMx2 domain. It happens to have the same magnitude of accumulation with that of TAPPI-PH (16%). We did not include the data regarding TAPPI-PH since we would like to focus on the basic aspect of PIP2/PIP3-mediated signaling in DNA damage repair and have deleted the reference to this in the text. The metabolism of PI(3,4)P2 and its role in DNA damage repair require further investigation.

2) *P9/ 10 – wortmannin is stated to have "caused an apparent increase in PIP2 and decrease in PIP3...that was reflected in the level of PIP2 and PIP3 PH binding domain accumulation". It is appreciated that the sentence contains the qualifier "apparent", but given that PPI levels were not measured, but are being inferred*

from the behaviour of the PH domain reporters, it would seem more accurate to say "as judged by the relative levels of accumulation of PIP2 and PIP3 PH binding domains".

A: The sentence was revised as following: "wortmannin caused an apparent increase in PIP2 and a decrease in PIP3 accumulation at damage sites, respectively, as judged by the relative levels of accumulation of PIP2- and PIP3-binding PH domains."

3) *P13 refers to experiments in Supp Fig. S4A, B showing that latrunculin A does not inhibit recruitment of ATM or DNA PK to sites of damage, however the figure actually contains the original western blots used in Fig. 4.*

A: The authors thank this reviewer for pointing out this editing mistake. The data showing an enhanced DNA-PK recruitment upon Lat A treatment was already displayed in Fig. 5I, while the same set of data for ATM staining was not shown since there was no significant difference. In this revised manuscript, the suppressed recruitment of endogenous ATR upon Lat A treatment was also included (Fig. 5H).

4) *A related point, the original blot documenting siRNA depletion of SF-1 is technically unconvincing. Better quality data is required to secure the important point that SF-1 was in fact successfully depleted.*

A: Fig. 4I was replaced with a better quality blots showing a clear depletion and rescue of SF1.

Reviewer #2 (Remarks to the Author):

This manuscript describes a novel role for nuclear phosphoinositide signalling in the DNA damage response. The role of nuclear PI signalling has remained enigmatic for many years, and this study shines a light on the role of PIP2 and PIP3 signalling in the nucleus and implicates IPMK in this process.

This study is well done, and definitely provides novel insights into the DNA damage response and nuclear PI signalling.

I have a few queries the authors should resolve before publication:

1) *Is the recruitment of phosphoinositides to sites of DNA damage dependent on active transcription? i.e. have they repeated their experiments with transcription elongation inhibitors such as DRB or splicing inhibitors. This is an important point, given that PIP2 localises to nuclear speckle domains, which are thought to be where certain mRNAs are processed prior to their export.*

A: The addition of transcription elongation inhibitor DRB at 25 μ M did not suppress the accumulation of NLS-tagged PLC δ PH domain at damage sites, suggesting that the damage response of PIP2 is independent of its role in transcription or mRNA processing machineries.

Fig. R1 The accumulation of NLS-tagged PLC δ PH was not affected by transcription elongation inhibitor, 5,6-Dichlorobenzimidazole 1- β -D-ribofuranoside (DRB). (A) The rapid accumulation of PLC δ PH at damages sites at the presence and absence of DRB at 25 μ M. (B) The accumulation magnitude of PLC δ PH of each cell after reaching plateaus at 30s.

2) *I do have a slight concern regarding their IPMK data. I am convinced that IPMK is required for the recruitment of PIPs from their rescue data and their data on ATM compensation following IPMK depletion is nice. However, at the time points they use (66h-72h), levels of HR proteins such as RAD51 and CHK1 (and maybe ATR) will be clearly decreased due to the effect of IPMK on nuclear export of HR mRNAs, so how can they be sure that the effects on ATR recruitment aren't due to this effect? Again, their rescue data are convincing, and the fact that it has a specific effect on Btk-PH and not PLC δ -PH is important. Nevertheless, a sentence should be*

included discussing this alternate possibility.

A: We agree that this could be an alternative explanation for suppressed ATR recruitment as seen in the immunofluorescent micrographs (Fig. 4M&N). Therefore, we checked for the expression of ATR using western blots. As shown in Fig.S3E, the level of ATR remained unchanged in cells transiently depleted with IPMK comparing to control cells using western blots.

3) *Quantitation - as far as I can tell, there is no indication of the number of independent experimental repeats performed for each experiment - For example in Figure 4: "Normalized quantification comparing the magnitude of recruitment of ATR (n=20), ATM (n=13), DNA-PKcs (n=14) and γ H2AX (n=47) within ROIs in control and IPMK-depleted conditions. Please refer to Supplementary Figure S3 for the representative images of ATM and DNA-PKcs staining. Error bars represent mean \pm S.E.M. in all panels." Is this from one experiment, or from multiple independent experiments?*

A: The number of repeats in addition to number of samples is now included. In the example of Figure 4, it was from three independent experiments. The cells were fixed within about 5 minutes after the first cells were microirradiated.

4) *They mention TSA data in the discussion but I can't seem to find the data in any figures : "The mechanism by which IPMK is activated for SF-1-PIP3 production remains elusive. A hint is provided by our results with cells pre-treated with a histone deacetylase (HDAC) inhibitor, TSA."*

A: We didn't include the data for TSA but only mentioned it at the end of the article, as we don't fully understand the mechanism by which TSA suppresses the recruitment of PIP2 and SF1. We included the TSA-related data here for reviewers' information.

Fig. R2 Histone deacetylase inhibitor, Trichostatin A, effectively suppressed the accumulation of both NLS-tagged PLCdPH-EGFP and EGFP-SF1 at damage sites. (A) Laser power-dependent accumulation of NLS-tagged PLCdPH-EGFP was nearly abolished to a baseline level at the presence of 5 μ M TSA. (B) The recruitment of EGFP-SF1 was also significantly suppressed by TSA at the same working concentration.

Reviewer #3 (Remarks to the Author):

This is an interesting article describing the accumulation of PIP(n) binding domains at sites of DNA damage in the nucleus and the impact of these domains on the ATR-dependent DNA damage response. Filamentous actin assembly is postulated to be an important factor in mediating PIP(n) driven ATRIP/ATR recruitment and Chk1 phosphorylation.

As presented, the data are convincing and controls using mutant lipid binding domains that cannot interact with their target lipid, and the lack of effect of these lipid-binding domains on other, parallel, DNA damage responses go a long way to argue against non-specific effects of the manipulations used. Further, the additional effects of reducing expression of IPMK and SF1 add weight to a nuclear-specific function of phosphoinositides in the ATR damage response pathway.

Overall, this manuscript has the potential to make an important advance in our understanding of the role of nuclear phosphoinositides but, I have several specific concerns that relate mostly to the interpretation of the results.

Specific comments:

1) Have the effects of non-nuclear-localised lipid binding proteins been investigated with respect to the DNA damage response ie are we

sure the effects of these domains are confined to the nucleus?

A: This question was addressed by comparing the ATR-GFP recruiting dynamics in the presence or absence of NLS-free PLC δ PH domain (See updated Fig. 2B&C). We showed in this result that the expression of cytoplasmic PLC δ PH domains did not affect the recruitment of ATR to the damage site, suggesting that these domains mainly take effect in suppressing ATR recruitment when they are expressed in the nucleus.

2) Further, have the effects of the nuclear-localised domains been investigated with respect to plasma membrane signalling e.g. AKT-phosphorylation?

We didn't investigate the effect of NLS-tagged PH domains with respect to Akt signaling. While the purpose of this experiment is to address whether Akt signaling in the plasma membrane is affected by blocking only the nuclear lipids, the fact that NLS-tagged PH domains were expressed both in the nucleus and in the cytosol defeated the purpose of this experiment (please refer to Fig. 1C or 2G-K for example).

However, there is reason to believe that sequestering nuclear PIP2 would lead to a reduced Akt phosphorylation. It was supported by the fact that the degree of Akt phosphorylation upon adding serum to serum-starved cells was significantly lower in IPMK-depleted cells comparing to control cells (Maag et. al., PNAS 2010). It should be noted that IPMK only localizes in the nucleus.

3) The effects of lipid binding domains selective for PI4P, PI45P2, PI34P2 and PIP3 suggest all of these lipids are involved in the ATR response. The effect of IPMK and SF1 then suggest phosphorylation of PI45P2 presented on SF1 is important. I find the conceptual framework here difficult to understand. Are the authors saying all of these lipids are linked within some sort of nuclear membrane-localised metabolic pathway (ie PI to PI4P to PI45P2), but then PI45P2 is extracted by SF1 and phosphorylated in this state to PIP3 by IPMK - but then what can be the involvement of PI34P2, since this is thought to originate via PIP3 5-dephosphorylation or PI4P 3-phosphorylation (by Class II PI3K)?

A: We would like to clarify two things regarding this issue. Firstly, we only mentioned in the text that the NLS-tagged TAPPI-PH domain was accumulated at the damage site with a magnitude of

increase at 16%, but didn't show whether the expression of NLS-TAPPI-PH impacted any impact on ATR-GFP recruitment. Therefore, it's actually not clear if all nuclear phosphoinositides including PI(3,4)P2 are involved in mediating DNA damage repair. Since we didn't show any relevant data about PI(3,4)P2, we took out the relevant discussion about TAPPI-PH domain to avoid confusion. So far, we showed that the turnover of lipids from PI4P -> PI(4,5)P2 -> PI(3,4,5)P3 was involved in mediating ATR recruitment. Our main effort was to unveil which nuclear lipid kinase is responsible for the phosphorylation of nuclear PIP2 to PIP3 and is also required in mediating ATR signaling.

Secondly, there is no membranous structure in the nucleus, as suggested by many electron microscopic studies. Our results certainly showed the importance of nuclear lipid-binding proteins such as SF1. SF1 belongs to a nuclear receptor family in which many members could bind to small molecules as their ligands. In fact, there is increasing evidence showing that nuclear PIP2/PIP3 serves as a ligand for some of these nuclear receptors. We proposed that the turnover from PIP2 to PIP3 happens on a protein-lipid complex (SF1-PIP2 complex) and which is a genuine substrate of IPMK. The turnover from PI4P to PI(4,5)P2 might also happen through a similar mechanism, but with different kinases.

- 4) *I'm also confused by the apparent efficacy of the PH domain probes to sequester the lipids – is the BTK-PH domain imagined to bind the head group of PIP3 whilst presented on SF1? The authors need to show this is possible (BTK-PH can't be acting prior to binding to SF1 otherwise there would be no necessity for IPMK phosphorylation of PI45P2 to PIP3 on SF1)? Or is the model here that PIP3 is released from SF1 into some sort of membrane compartment before it acts to regulate actin and ATR, and it is at this point that the BTK-PH can interfere?*

A: Currently there is no co-crystal structure for an SF1-PIP3-BtkPH complex, but one can infer from the simulated structure based on the crystal structure of individual components. The Blind group has presented simulated structures showing the formation SF1-PIP3-PDK1-PH complex based on the reported crystal structure of both SF1-PIP3 and PDK1-PH domain, separately (Fig. R2B). While PDK1-PH and Btk-PH share high structural similarity (Fig. R2A), we assume that BtkPH is capable of recognizing the exposed headgroup of PIP3 while bound to SF1.

Fig. R3 The structural comparison of PIP3-binding PH domains and simulated structure of SF1-PIP3-PDK1-PH complex(A) Structural alignment of Btk-PH (gold) vs. PDK1-PH (cyan) together with IP4 as a ligand. BtkPH is structurally very similar to PDK1-PH domain. (B) Docking simulation of the PH domain of PDK1 (cyan) interacting with SF1 (gold)/ PIP3 (stick structure), showing possible complementarity between two independent crystal structures. The head group (cyan) is from the PDK1 structure whereas the bridging phosphate and acyl chains are from the SF1-PIP3 complex. (Panel B adapted from Blind et. al., PNAS 2014, 111(42), 15054-9.)

5) *I can understand why the anti-PIP2 antibody would present a different overall image to that of the PI45P2-selective PH domain, but wouldn't you anticipate some level of co-localisation if they are both binding to the same pools of PI45P2? I couldn't discern any evidence of this.*

The co-localization of nuclear PIP2 speckles with PLC δ PH domain in the nucleus is compromised due to the high expression level of PH domain, which largely decreases the signal-to-noise ratio to identify PIP2-enriched fine structures. In order to show that anti-PIP2 speckles are indeed also recognized by NLS-tagged PLC δ PH domains, a new supplementary figure S4 was added. In this figure, the nucleus of an NLS-PLC δ PH-expressing cell was fixed after being exposed to UV for 10min and recovered for an hour, and then stained for anti-PIP2 antibody and anti- γ H2AX antibody. The line profile clearly demonstrated that anti-PIP2 speckles highly co-localized with anti- γ H2AX foci. Further, PLC δ PH domain was also locally concentrated around the anti-PIP2 speckles as indicated by the line profile, although with a much lower S/N ratio. Visually, it only becomes clearly if one greatly enhances the contrast. It's not

discernable otherwise.

6) *Further, in Figure 1, wouldn't you anticipate that expression of the PI45P2-selective PH domain would reduce the number of anti-PIP2 puncta in the no-UV condition as well as 1hr after UV irradiation?*

A: We didn't anticipate that expression of PLC δ PH would reduce the number of anti-PIP2 puncta. Instead, we anticipated a reduced signal intensity of anti-PIP2 puncta, which is particularly evident when comparing the brightness of anti-PIP2 puncta of panel A compared to B after irradiation with UV.

7) *The manuscript would be strengthened by a clear, quantitative defect in the ATR-dependent DNA damage repair response in IPMK-KO cells.*

A: To generate an IPMK-KO stable MEF cell line is possible as reported by the Resnick group (Maag et. al., PNAS 2010). A prolonged IPMK depletion was reported to reduce the mRNA export and the production of homologous recombination proteins, which leads to genome instability (Wickramasinghe et. al., Mol. Cell 2013). Therefore, we would argue that results with a stable cell line of IPMK-KO cells would be difficult to interpret compared to a transient knock down.

8) *The cartoon in Fig 4L is not helpful without depicting the conversion of PI45P2 to PIP3.*

A: The cartoon in Fig.4L (now Fig.4K) was modified accordingly.

Reviewer #4 (Remarks to the Author):

I have enjoyed reviewing this article. The experimentation is creative and yields novel insights.

In this article Wang et al report on nuclear phosphoinositide signaling following DNA damage afflicted using a UV laser (micro irradiation). The work relies on PH-domain containing peptides to recognize specific classes of phosphoinositides. These PH-domain containing peptides serve to both deplete and (because they are fluorescently tagged) quantify and localize specific phosphoinositides. Most of the work is done in a cell line where nucleolar accumulation of the PH-domain containing peptides complicates interpretation of results. However, the authors recognize this complication and select regions of interests (ROIs) outside of the nucleoli. The major finding is that there are two forms of DNA damage repair, one reliant on PIP signaling, resulting in recruitment of ATR to sites of DNA damage, the other one, resulting in

recruitment of ATM and DNA-PK, is independent of PIP signaling. The technology is sophisticated and quite indirect (the intensity of fluorescent PH-domain containing peptide is taken as a read-out for PIP concentration in a given ROI). With that comes a risk of over interpretation of the observations. The data presentation requires clarifications:

1) Fig. 1 How is 'normalized mean intensity' measured and calculated. The segment on image quantification in the methods section only describes how ROIs re evaluated. But here the entire nucleus was evaluated. How was that done? Is green fluorescence of the entire nucleus measured, are speckles counted? By eye, the difference between the PH-domain transfected cells and the control is not different when looking at the PIP2 immunization. (A versus C). However the bar graph does show a significant difference.

A: We didn't quantify the PIP2 speckles and measured the mean fluorescence intensity of the whole nucleus instead, as we didn't make presumptions regarding the distribution of nuclear PIP2 in response to DNA damage. The "normalized mean intensity" in Fig.1 was done by measuring the mean fluorescence intensity within each nucleus, followed by background subtraction, averaging, then normalized against the same value measured from non-UV treated, non-transfected cells. A paragraph for clarification is now included in the material and methods section.

From our inspection, the difference in the staining of PIP2 between Fig. 1A&C was clear enough to catch by eye both on the screen and on printed paper.

Fig. 2 Convincingly shows specificity of the PH-domain containing peptides for certain PIPs, and failure to recruit ATR pathway members to sites of damage

2) Fig. 3 Failure of PH-domains to accumulate at sites of DNA damage in dead cells could be due to any number of reasons, they do not proof active lipid synthesis as the authors claim. (I-K) The difference in ROI fluorescence intensity is a mere 5% for the BTK-PH domain and increase for the PLC-delta PH domain likewise. While the results are statistically significant, this raises the question if they are biologically significant. Is ATR recruitment to sites of micro irradiation wortmannin-sensitive?

A: As the reviewer pointed out, the failure of PH-domains to

accumulate at DNA damage sites could be due to many other reasons, this data only serves as part of the evidence to support our claim. In fact, we only claimed that the accumulation of PH domain would be an active process from this experiment. The claim of active lipid synthesis came after we also demonstrated that PH domain accumulation is ATP-dependent and wortmannin-sensitive, and PIP2 was concentrated at damage sites as revealed by immunostaining. The second question is regarding the biological significance of wortmannin-induced change in BtkPH accumulation magnitude. We therefore performed experiments as proposed by this reviewer. We demonstrated that the recruitment of ATR was indeed suppressed by wortmannin in a dosage-dependent manner (Fig. 5C, F&G).

3) *Fig. 4 E->J Similar issue. %Inc (percent increase I presume) is the major read-out, and while statistically significant, one wonders about its biological meaning.*

A: The biological significance of the measurements in Fig. 4E to J is also given by western blotting experiments proposed by the first reviewer (see the comments from first reviewer, major comment 2). Our new western blot data clearly demonstrated that the depletion of IPMK and SF1 significantly suppressed Chk1 phosphorylation. While we proposed that nuclear PIP2 and its phosphorylation serves as the link between IPMK, SF1 and Chk1 activation, the biological meaning of these small changes in accumulation magnitude in the ATR signaling is self-evident.

4) *Fig. 6 Don't understand subpanel D, legend too cryptic*

A: It was revised as the following: "The halftime measured from panel C and the halftime from complementary FRAP measurements of the three NLS-tagged proteins showed that all three proteins were equally diffusive in the nucleus but were recruited to the damage sites with different time constants.

Statistics:

Method section needs a separate statistics section that explains how many experimental repeats were done, how many cells per experiments counted, justification of those numbers, how normalization was done.

A: Additional statistic session is added in the main text.

Use of the standard error of the means (SEM) rather than Standard Deviation (SD) not explained when presented such as in Fig. 4, 5 etc. Throughout the figures statistical tests need to be added and p-values presented, this is not done consistently. Also number of repeats of independent experiments (not only number of cells analyzed per experiment) need to be indicated.

A: P-values for Fig. 1 and Fig. S1 are now added. The ranges of p-values are indicated as number of stars as explained in the figure legends. The actual numbers of p-values are not given as the layout is crowded. Number of repeats of independent experiments is indicated in all figure legends.

Cell line: Need to explain why results were only obtained in U2OS osteosarcoma cells with the exception of MEFs in one experiment. If this mechanism is true in general, it should be shown in a range of cell lines. Is there any reason to assume that differs between normal and cancer cells?

A: We didn't assume that there is a difference between normal and cancer cells. In fact, the two cell lines seem to give similar results in the key experiments. Most laser microirradiation experiments were done in U2OS because its flat and large nucleus makes it easier for observation. We used both U2OS and MEF cells for knock down because some of the knock downs in U2OS cells using siRNA were not successful; therefore, we turned to use shRNA against mouse proteins in MEF cells where we could use antibiotic selection to improve KD efficiency.

The text needs revision of the language. It is in general too wordy, and too many acronyms are used, it is hard to follow.

The discussion fails to explain what this newly discovered mechanism might mean for normal or cancerous cells, how it might have evolved, what the significance of the differential regulation of ATM/DNA-PK on the one and ATR recruitment on the other might be, and why the timing which the authors delineate, might be of importance.

As detailed above, the paper needs major revisions and is not acceptable for publication in its current form.

We improved the text and added further explanation about the significance of the ATR pathway.

REVIEWERS' COMMENTS:

Reviewer #1 (Remarks to the Author):

Wang and colleagues have significantly strengthened their manuscript, particularly through confirmation of key observations via western blotting.

Minor points

1) Lines 268/ 269/ Figure 4 – references to panels K and L seem to be reversed.

2) Lines 343-346 – some additional brief description of the rationale for using mDia2 as a marker for formin recruitment would be useful for those unfamiliar with the actin field.

Reviewer #2 (Remarks to the Author):

The authors have addressed all of my concerns, and I recommend publication. Congratulations on an interesting study!

Reviewer #3 (Remarks to the Author):

I have no further comments that I think would lead to an improved paper.

Reviewer #4 (Remarks to the Author):

This revision is in many technical ways responsive to the critique, but it remains a hard-to-read manuscript. It is very technical, and unfortunately, the language as well as the labelling of the figures is often inconsistent, which makes it exceedingly hard to figure out the details.

The language has not improved sufficiently. The language shifts back and forth between past and present tense, and there are many incomplete and incoherent sentences.

'ionic' radiation - I suppose 'ionizing' is meant?

'As hypothesized, both NLS-tagged mDia2 and Utr230 accumulated at damage sites after NLS-PH domains' - what does that mean? Temporal sequence? Can this really be concluded from Fig. 6B?

line 258:

'25°C or after ATP depletion for 30' 30 what?? minutes? hours? seconds?

line 347 'While the half-time for mDia2 accumulation was about 2 seconds, the accumulation of Utr230 followed a double exponential rise with a fast component that leveled off at 100% of background fluorescence intensity (diffusion of Utr230) and a slow component showing accumulation above background with a half-time about 4 seconds' That is hard to understand. All curves in 6 C are biphasic. If the functions derived from the experimental data are really fundamentally different, please provide those functions and explain properly.

'mDia' is never introduced. Why is this being used.

Methods are still not really clear: 'The photobleaching time was limited at 0.25 s to avoid significant diffusive recovery and recovery halftimes were determined using standard techniques following a double normalization method.'

What is meant by 'a double normalization method'? Please provide either reference or spell out how you did this.

Figure 6 E: 'FRAP' is opposed to 'DNA damage'. Are we observing fluorescence recovery after two different insults to the nucleus, bleaching versus UV-DNA damage radiation?

The tags mDia and Utr230 are known to be small molecules that bind whatever they bind reversibly, not covalently, why do a negative control experiment with FRAP? As the authors point out, FRAP only informs about the mobility of the reporter, and does not allow any conclusions on nuclear actin in this context.

As initially pointed out, this is a sophisticated methodology and the insights are potentially important and intriguing. My technical concerns have been adequately addressed.

REVIEWERS' COMMENTS:

Reviewer #1 (Remarks to the Author):

Wang and colleagues have significantly strengthened their manuscript, particularly through confirmation of key observations via western blotting.

Minor points

1) Lines 268/ 269/ Figure 4 – references to panels K and L seem to be reversed

The mislabeling is corrected.

2) Lines 343-346 – some additional brief description of the rationale for using mDia2 as a marker for formin recruitment would be useful for those unfamiliar with the actin field.

Due to the constraint of space, the rationale of investigating mDia2 recruitment was briefly explained in the following sentence: “Since PPIs were implicated in cytoplasmic actin filament polymerization by regulating the localization of formins such as mDia1 and mDia2 through interacting with their N-terminal base domains, it was logical to test if nuclear PPIs promoted damage-induced nuclear actin assembly and whether such actin assembly was catalyzed by recruiting formins to damage sites.” -P.12

Reviewer #2 (Remarks to the Author):

The authors have addressed all of my concerns, and I recommend publication. Congratulations on an interesting study!

Reviewer #3 (Remarks to the Author):

I have no further comments that I think would lead to an improved paper.

Reviewer #4 (Remarks to the Author):

This revision is in many technical ways responsive to the critique, but it remains a hard-to-read manuscript. It is very technical, and unfortunately, the language as well as the labeling of the figures are often inconsistent, which makes it exceedingly hard to figure out the details.

The language has not improved sufficiently. The language shifts back and forth between past and present tense, and there are many incomplete and incoherent sentences.

We have revised it again and paid attention to the tense used. Tenses of a few sentences are corrected.

'ionic' radiation - I suppose 'ionizing' is meant?

Yes, this typo is corrected.

'As hypothesized, both NLS-tagged mDia2 and Utr230 accumulated at damage sites after NLS-PH domains' - what does that mean? Temporal sequence? Can this really be concluded from Fig. 6B?

This result only suggests that these components were recruited with different halftimes following a temporal sequence: PH domain → mDia2 → Utr230. Whether there is a causal link within this recruiting sequence cannot be known but requires other experiments. To reduce confusion, we rephrased the sentence as follows:

“As hypothesized, the halftimes for recruiting both NLS-tagged mDia2 and Utr230 to damage sites were longer than that of the NLS-PH domains (Figure 6c,e).”

line 258:

'25°C or after ATP depletion for 30' 30 what?? minutes? hours? seconds?

It should be in minutes. The missing unit is added back.

line 347 'While the half-time for mDia2 accumulation was about 2 seconds, the accumulation of Utr230 followed a double exponential rise with a fast component that leveled off at 100% of background fluorescence intensity (diffusion of Utr230) and a slow component showing accumulation above background with a half-time about 4 seconds' That is hard to understand. All curves in 6 C are biphasic. If the functions derived from the experimental data are really fundamentally different, please provide those functions and explain properly.

All three curves were fitted with a double exponential curve initially. The fast component is the fluorescence recovery due to free diffusion of the probes, while the slow component is due to the active accumulation of the probes. The fast and slow components of the PH domain are not resolvable and we therefore fit the curve for PH domain with a single exponential rise instead. Both mDia2 and Utr230 followed a double exponential rise with similar fast components that leveled off at 100% fluorescence intensity. The sentences are modified as the following for clarification:

“While the curve for PLC δ PH was fitted properly with a single exponential rise, the curves of mDia2 and Utr230 recovery followed a double exponential rise with the fast component leveling off at 100% fluorescence intensity. The halftimes of the slower components corresponded to halftimes of 2 and 4 seconds, respectively.”

'mDia' is never introduced. Why is this being used.

Formins such as mDia1&2 possess a basic domain at their N-terminus, which allows them to interact with negatively charged lipids such as PS and PIP2. To explain why mDia2 is used in this manuscript, a few sentences were added:

“Since PPIs were implicated in cytoplasmic actin filament polymerization by regulating the localization of formins such as mDia1 and mDia2, it was logical to test if nuclear PPIs promoted damage-induced nuclear actin assembly and whether nuclear actin polymerization was catalyzed by recruiting formins to damage sites.” -P.12

Methods are still not really clear: 'The photobleaching time was limited at 0.25 s to avoid significant diffusive recovery and recovery halftimes were determined using standard techniques following a double normalization method.'

What is meant by 'a double normalization method'? Please provide either reference or spell out how you did this.

Double normalization refers to a two-step normalization process proposed by Phair and co-workers in 2003¹. In this procedure, the data was first normalized against the pre-bleaching intensity and then against nearby non-bleached nucleus to correct for fluorescent loss. It's now a commonly used procedure for FRAP data. Additional normalization, which set the first post-bleaching data point to zero, is optional and is not adopted in our analysis.

Figure 6 E: 'FRAP' is opposed to 'DNA damage'. Are we observing fluorescence recovery after two different insults to the nucleus, bleaching versus UV-DNA damage radiation?

The tags mDia and Utr230 are known to be small molecules that bind whatever they bind reversibly, not covalently, why do a negative control experiment with FRAP? As the authors point out, FRAP only informs about the mobility of the reporter, and does not allow any conclusions on nuclear actin in this context.

FRAP measurements were not meant to serve as a negative control for laser microirradiation. The major negative controls for both mDia2 and Utr230 were still EGFP-3xNLS as shown in Figure 3B. (Another negative control for Utr230 was the addition of LatA). While three different probes were recruited to the damage site with different time constants, we inferred from FRAP that the diffusivity of the three probes were roughly the same. Together, these results indicated that the difference in recruiting dynamics resulted from the temporal sequence of the accumulation instead of the diffusivity of the reporters.

As initially pointed out, this is a sophisticated methodology and the insights are potentially important and intriguing. My technical concerns have been adequately addressed.

Reference

1. Phair RD, Gorski SA, Misteli T. Measurement of Dynamic Protein Binding to Chromatin In Vivo, Using Photobleaching Microscopy. **375**, 393-414 (2003).